# The *Trypanosoma cruzi* Antigen and Epitope Atlas: antibody specificities in Chagas disease patients across the Americas

Alejandro D. Ricci [1,2,12], Leonel Bracco [1,2,12], Emir Salas-Sarduy [1,2], Janine M. Ramsey[3], Melissa S. Nolan[4], M. Katie Lynn [4], Jaime Altcheh[5,6], Griselda E. Ballering[5], Faustino Torrico[7], Norival Kesper[8], Juan C. Villar[9], Iván S. Marcipar[10], Jorge D. Marco [11] & Fernán Agüero [1,2] ✉

During an infection the immune system produces pathogen-specific antibodies. These antibody repertoires become specific to the history of infections and represent a rich source of diagnostic markers. However, the specificities of these antibodies are mostly unknown. Here, using high-density peptide arrays we examined the human antibody repertoires of Chagas disease patients. Chagas disease is a neglected disease caused by *Trypanosoma cruzi*, a protozoan parasite that evades immune mediated elimination and mounts long-lasting chronic infections. We describe a proteome-wide search for antigens, characterised their linear epitopes, and show their reactivity on 71 individuals from diverse human populations. Using single-residue mutagenesis we revealed the core functional residues for 232 of these epitopes. Finally, we show the diagnostic performance of identified antigens on challenging samples. These datasets enable the study of the Chagas antibody repertoire at an unprecedented depth and granularity, while also providing a rich source of serological biomarkers.

Although the cellular and molecular mechanisms that produce diverse antibody repertoires and underpin antibody affinity maturation in response to infection are largely understood[1–4], a comprehensive description of their specificities in different infected individuals has been hindered by the lack of powerful tools.

Synthetic peptides have been used historically to map continuous antibody-binding epitopes or to find key residues for protein binding at a small scale[5–7]. With the introduction of peptide arrays, it was possible to display large numbers of peptides on a solid surface at addressable positions[8,9]. Given the sustained increase in the densities achieved by the in situ synthesis of peptides using maskless photolithography[10–15], it is now possible to display complete proteomes in a single slide[16,17], opening the door to high-throughput serological screenings.

[1]Instituto de Investigaciones Biotecnológicas (IIB) — Consejo Nacional de Investigaciones Científicas y Técnicas (CONICET), San Martín, Buenos Aires, Argentina. [2]Escuela de Bio y Nanotecnologías (EByN), Universidad de San Martín (UNSAM), San Martín, Buenos Aires, Argentina. [3]Centro Regional de Investigación en Salud Pública, Instituto Nacional de Salud Pública, Tapachula, México. [4]Laboratory of Vector-borne and Zoonotic Diseases, Arnold School of Public Health, University of South Carolina, Columbia, SC, USA. [5]Hospital de Niños "Ricardo Gutierrez", Ciudad Autónoma de Buenos Aires, Buenos Aires, Argentina. [6]Instituto Multidisciplinario de Investigaciones en Patologías Pediátricas (IMIPP) — GCBA-CONICET, Buenos Aires, Argentina. [7]Fundación CEADES, Cochabamba, Bolivia. [8]LIM-49, Hospital das Clínicas HCFMUSP, Faculdade de Medicina, Universidade de São Paulo, São Paulo, Brasil. [9]Facultad de Ciencias de la Salud, Universidad Autónoma de Bucaramanga y Fundación Cardioinfantil — Instituto de Cardiología, Bogotá, Colombia. [10]Facultad de Ciencias Médicas y Facultad de Ciencias Biológicas, Universidad Nacional del Litoral, Santa Fe, Argentina. [11]Instituto de Patología Experimental, Universidad Nacional de Salta — Consejo Nacional de Investigaciones Científicas y Técnicas (CONICET), Salta, Argentina. [12]These authors contributed equally: Alejandro D. Ricci, Leonel Bracco. ✉e-mail: fernan@iib.unsam.edu.ar

Chagas disease, also known as American trypanosomiasis, is a lifelong infection caused by the protozoan parasite *Trypanosoma cruzi*. Despite being discovered ~100 years ago, the condition remains a major social and public health problem in Latin America and is regarded as a neglected tropical disease by the World Health Organization[18].

After initial infection, the parasite evades immune-mediated elimination and mounts long-lasting chronic intracellular infections. Due to the low parasitemia observed during the chronic stage of the disease, serological methods are the preferred choice for the diagnosis of infection. Although available diagnostic tests give satisfactory results in most cases, there is currently no reference ('gold') standard for the diagnosis of infection, hence discordant results remain a possible cause of undetected cases[19–21]. Serological discordance is particularly high in human populations in North America[20,22–24]. Also, there are urgent needs to improve or fill vacant niches with customised serological tools and assays to monitor existing treatments or clinical trials[25–27] and to detect the early onset of Chagas disease pathology[28], both in active case finding and management and in epidemiological and disease surveillance programmes[29,30].

In this article, we report on the comprehensive survey and characterisation of the human antibody responses and specificities against *T. cruzi* using state-of-the-art high-density peptide arrays. This survey investigated the antigenicity of the predicted proteomes of two *T. cruzi* strains in 71 Chagas disease subjects from diverse human populations across the Americas. As a result, we produced a comprehensive atlas of antigens and their linear epitopes, providing a unique resource for understanding adaptive immune responses against this parasite. We also validate identified antigens and show their impact on the diagnosis of challenging samples.

## Results

### CHAGASTOPE-v1: design of a high-density peptide array for antigen and epitope discovery

We used high-density peptide arrays to perform a high-resolution antigen discovery screening and epitope mapping for complete *T. cruzi* proteomes. We designed an array that included protein sequences encoded in the genomes of two *T. cruzi* strains: the genome reference CL-Brener strain (19,668 proteins, Discrete Typing Unit (DTU) TcVI, hybrid)[31], and the Sylvio X10 strain (10,832 proteins, DTU TcI, non-hybrid)[32]. The selection criteria considered the epidemiological relevance of these representative lineages in the context of Chagas disease, and the fact that TcVI strains are hybrids of ancestral DTUs TcII and TcIII[33,34], resulting in most of its genes being represented in the genome by their two ancestral allelic versions[31,35]. Therefore, by using strains from DTUs TcI and TcVI we maximised the display of relevant peptide variants with only two genomes.

Based on this analysis and the high-density peptide array capacity, we produced a tiling display of 30,500 *T. cruzi* proteins using 16mer peptides with an offset of 4 amino acids (overlap of 12 residues between consecutive peptides). The resulting array containing 2,441,908 unique peptides was named CHAGASTOPE-v1 and was used for the discovery screening (see Fig. 1). Additional details on the contents of the CHAGASTOPE-v1 array design are available in the Methods and in Supplementary Data S1–S3.

### Discovery screening: a high-content serological screening reveals diverse antibody repertoires in Chagas disease

Using the CHAGASTOPE-v1 array design, we performed a discovery screening using pooled serum samples from positive donors and paired negative sample pools from healthy subjects for six different geographical regions across the Americas (Fig. 1). In total, we profiled 12 different pooled serum samples, 6 from infected subjects and 6 from healthy subjects (see Supplementary Data S2 for details). To test for cross-reacting epitopes, two additional pools were profiled: a pool from leishmaniasis-positive individuals and a matched pool from

leishmaniasis-negative (also Chagas-negative) samples from the same geographic area. All 14 samples were assayed in duplicate, and all technical replicates had high signal correlation (see Supplementary Fig. S1).

After normalising the antibody-binding fluorescence signal across experiments, we reconstructed each original protein sequence, used consecutive peptides to perform signal smoothing (to remove outliers, see Methods) and generated visualisations of the antibody-binding signal for each protein and for each assayed sample (see example in Fig. 2A and the full list in Supplementary File S1).

To identify the more reactive proteins, we compared signals obtained across experiments with pools of Chagas-negative and Chagas-positive subjects and defined an antigenicity signal threshold. We chose a very conservative threshold of 10,784.80 fluorescence units (the statistical mode plus 4 standard deviations). Any group of two or more consecutive peptides above this threshold was defined as an antibody-binding peak (see Fig. 2), resulting in 18,199 peaks for the Chagas-positive subjects. We also observed 3644 reactive peaks in Chagas-negative subjects (see Supplementary Fig. S2B and Supplementary Data S4). After removing these non-specific peaks, we obtained 16,737 Chagas-specific antigenic peaks across 7707 proteins.

Because some peaks were either close or partially overlapping with one another (in the same analysed sample or across different samples, see Fig. 2C), we combined neighbouring peaks into non-overlapping antigenic regions. This resulted in 9547 antigenic regions across both proteomes (see Methods). Furthermore, because the analysed *T. cruzi* genomes have several large gene families, a significant number of reactive regions displayed evident sequence similarity amongst them. Hence, we grouped these antigenic regions into 3868 non-redundant clusters based on sequence similarity using protein BLAST (see Supplementary Fig. S2C and Methods). The identification of reactive peptides, peaks and regions, and their cognate antigenic proteins is summarised in Table 1.

### The immune responses in Chagas disease subjects are highly diverse

The complete map of measured antibody-binding reactivities across pooled samples provided a broad view of the diversity of the antibody repertoire developed in response to *T. cruzi* infections. We next analysed how reactive regions were shared amongst pooled samples and observed a large set of non-shared reactive epitopes in each sample (see Fig. 3). These were not technical artefacts as they were reproducibly identified in the technical replicates and were also supported by the reactivity of overlapping neighbouring peptides (pseudo-replicates within each experiment).

This set of non-shared (private for pool) antigenic regions was followed by a long tail of shared epitopes with increasing seroprevalence. This observation suggests that the antibody response in Chagas disease is derived from a large and diverse set of antibody-producing B-cell clones (no sera pools shared more than 30% of its antigenic peptides, see Supplementary Fig. S2A).

Particularly important for serology-based applications is the large set of shared clusters of antigenic regions across Chagas disease subjects (166 shared by at least 4 of the analysed pooled samples), including 43 clusters that were reactive in all samples. The seroprevalence for all 3868 antigenic clusters can be found in Supplementary Data S5, while the full list of 9547 antigenic regions with all their details can be found in Supplementary File S2.

We also screened the same array design against a pool of samples from leishmaniasis-positive individuals to identify cross-reacting epitopes (see Supplementary Data S6). Overall, there was very low cross-reactivity of *T. cruzi* peptides against this pool. Out of the 3868 clusters of antigenic regions, only 104 (2.7%) had reactivity against this leishmaniasis sample (using the same threshold used for *T. cruzi* samples).

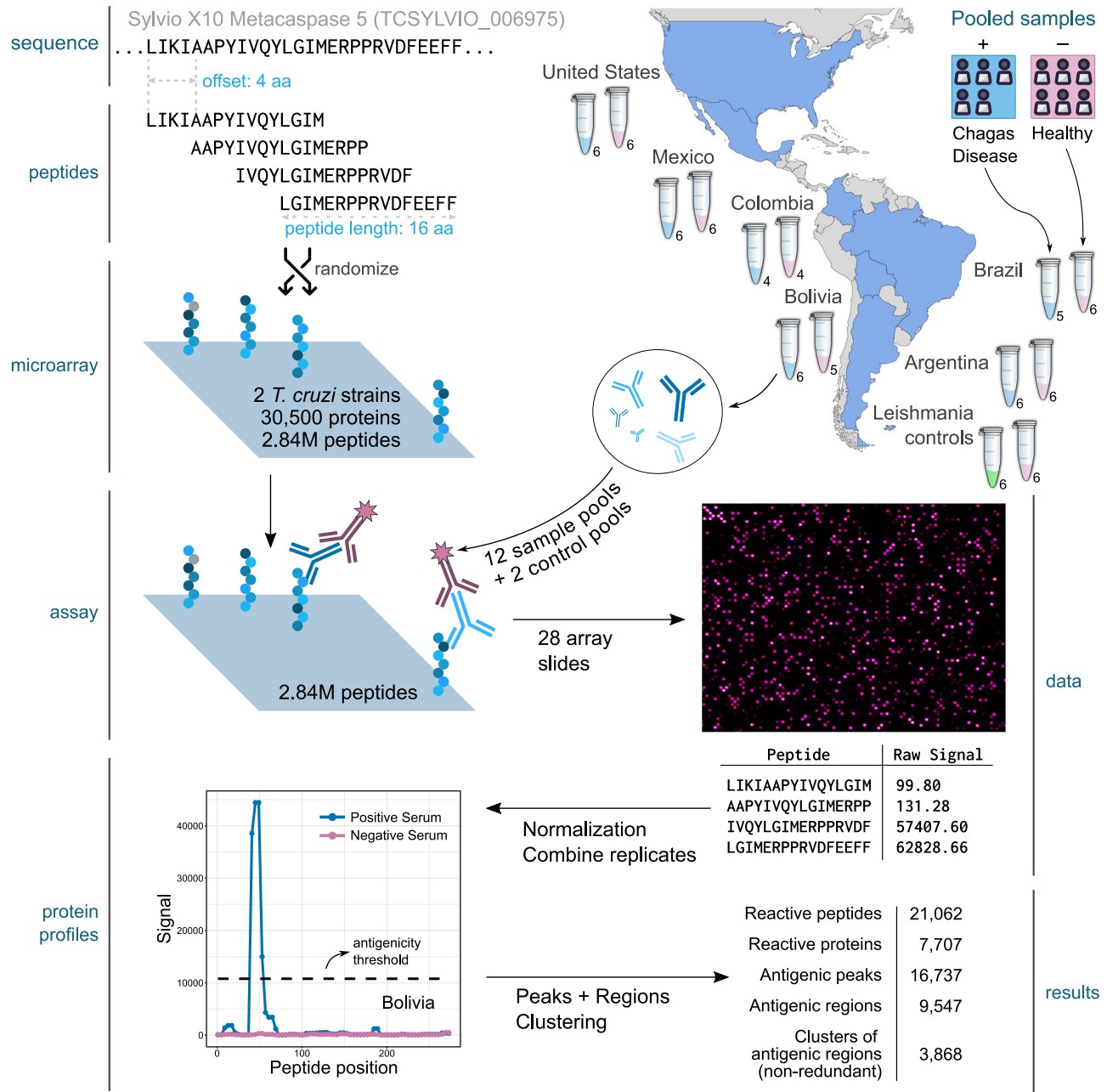

**Fig. 1 | Summary of the discovery screening.** The figure shows a schematic representation of the steps followed to analyse two *T. cruzi* proteomes (CL-Brener and Sylvio X10) using pooled serum samples across the Americas (one pool from Chagas-infected individuals and one from healthy subjects from Argentina, Brazil, Bolivia, Colombia, Mexico, and the United States). The numbers below each tube represent the number of individual sera in each pool. An additional pooled serum sample of Leishmania-infected individuals and their healthy counterparts were used to study cross-reactivity. The protein used for this example is the metacaspase 5 protein from Sylvio X10 (TCSYLVIO_006975), which has 291 residues and was represented in each array using 69 peptides (16mers with an offset of 4 aa), of which only the first 4 peptides are shown here. Third-party image elements: the coloured map was created with MapChart, tubes and person open-licensed clipart are from Pixabay and SVG Repo.

This included 5 of the 43 clusters that had reactivity in all Chagas-positive samples, although for 3 of these, only a small percentage of the regions in the cluster were cross-reactive.

### Individual patient resolution provides insights into the diagnostic value of identified antigens and epitopes

The previous screening provided ample information on the diversity and specificities of the antibody repertoire of Chagas disease patients. However, the use of pooled samples limited the analysis of the data. Next, we increased the epitope mapping resolution and assessed the reactivity of epitopes for individual patient samples.

Because 99% of peptides showed no antibody-binding at the screening stage, we removed these from the new design to focus only on the antigenic regions. Working with these smaller regions of the proteome allowed us to increase the epitope mapping resolution, using 16mers with an overlap of 15 residues between consecutive peptides.

This second peptide design (named CHAGASTOPE-v2) included peptides from the 9,547 protein regions that were both antigenic (reactive with at least one Chagas-positive sample) and that showed no signal from healthy control subjects. To ensure that entire reactive peaks in each region were included in the design, we included up to 32

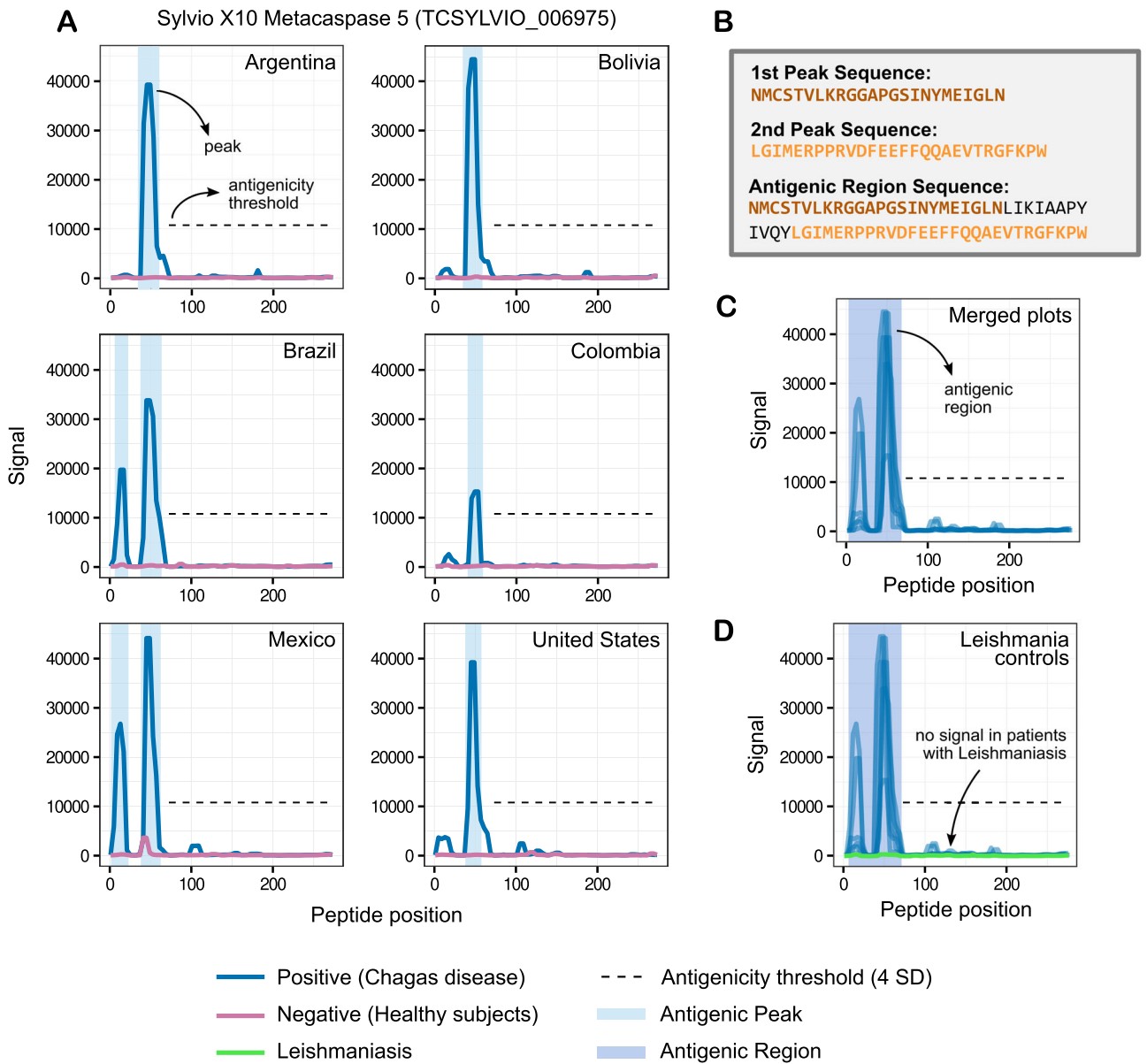

**Fig. 2 | Antibody-binding profiles, peaks, and regions.** The normalised fluorescence signal of each peptide in any given protein was used to produce the antibody-binding profiles for an antigen. The *y*-axis shows fluorescence units. The *x*-axis shows peptide positions along the protein sequence. Each subplot was produced using data from 4 high-density peptide arrays (2 replicas for Chagas disease subjects and 2 for matched healthy subjects, see main text). The figure serves to illustrate how we defined peaks (groups of consecutive peptides over the signal threshold) and antigenic regions (groups of neighbouring peaks). **A** Reactivity subplots for different sera pools for the Sylvio X10 metacaspase 5 protein. The antibody-binding profiles are shown in blue for the Chagas-positive sample pools (infected) and in magenta for the Chagas-negative pools (healthy). We used a conservative 4 SD antigenicity threshold throughout the discovery phase, which can be seen as a black dashed line. **B** Peptide sequences of the reactive peaks and regions in (**A**). **C** Reactivity plot merging data from all sample pools. **D** Same as (**C**) and showing the signal for the leishmaniasis-positive serum samples (in green).

---

additional peptides from the surrounding non-reactive borders of each region (16 from each side). This contributed to the v2 array design with 241,772 unique peptides from this set (see Methods for details).

To expand the resolution of epitopes down to individual patients, we used sectorised high-density peptide arrays to assay more samples (primary antibodies) in parallel in the same slide (see Supplementary Data S1). In these 12-plex arrays, the same set of CHAGASTOPE-v2 peptides was replicated in each sector. A total of 12 CHAGASTOPE-v2 arrays (144 sectors) were used to analyse 71 individual serum samples in duplicate, 33 of which were part of the pools analysed in the previous step. A set of 38 additional serum samples were analysed from other Chagas-positive subjects from the same 6 geographic regions (see Supplementary Data S2). The replicas showed high Pearson correlation coefficients (>0.8, see Supplementary Fig. S3). Because these arrays have a much lower content of non-reactive peptides, we recalculated the antigenicity threshold, resulting in a threshold of 5814.81 relative fluorescence units (statistical mode plus 2.4 standard deviations, see Methods).

The resolution of antibody-binding reactivity at an individual level provides information on the seroprevalence of each antigenic region and led us to depict different antibody-binding profiles across subjects. Figure 4A shows an antigen with a single antigenic region (reticulon domain-containing protein, TCSYLVIO_003288). The observed seroprevalence at the level of the whole protein is aligned with the seroprevalence observed at the single-epitope level. Figure 4B shows a different extreme example, the hypothetical protein encoded by gene

## Table 1 | Discovery screening finding summary

|  | CL-Brener | Sylvio X10 | Pangenome |
|---|---|---|---|
| Total proteins | 19,668 | 10,832 | 30,500 |
| Total peptides (non-redundant) | 1,809,351 | 1,187,499 | 2,441,908 |
| Reactive peptides (non-redundant) | 16,435 | 7709 | 21,062 |
| Antigenic peaks | 12,720 | 4017 | 16,737 |
| Antigenic regions | 6710 | 2837 | 9547 |
| Reactive proteins | 5328 | 2379 | 7707 |
| Clusters of antigenic regions (non-redundant) | – | – | 3868 |

The table displays the number of reactive peptides, peaks, regions and proteins found in the primary screening. When analysing the number of peptides, each unique sequence was only counted once.

TCSYLVIO_005669, another antigen reported in this work. This protein displayed a contiguous antigenic region composed of 7 different antibody-binding peaks (epitopes), each with a unique set of signal and seroprevalence characteristics. In this instance, the global seroprevalence for this antigen was not aligned with the seroprevalence of individual epitopes. Finally, Fig. 4C shows the repetitive antibody-binding profile of the Ag36/MAP antigen[36] for US samples. Resolution of how antibodies from different individuals recognise this antigen provides insights to improve diagnostic reagents. As shown in the figure, while most individuals display reactivity for all overlapped epitopes in this repeat, some individuals (e.g., US_P3, US_E2) have antibodies with different binding preferences along this antigenic repetitive unit.

Supplementary Files S3 and S4 contain the antibody-binding profiles for all antigens and all analysed patient samples tested in CHAGASTOPE-v2 (plots similar to those in Fig. 4). This is a rich dataset

for future studies on the serology and immune responses of Chagas disease patients and serves as a reference for other infectious diseases.

Resolution of antibody-binding to defined peptides in each subject also allowed us to explore reactivity against *T. cruzi* strains. Figure 5 summarises the reactivity of all 71 subjects against peptides found exclusively in the CL-Brener or in Sylvio X10 strains. Subjects displayed reactivity to an average of 3908 CL-Brener exclusive peptides (min: 2606; max: 5113, std: 445) and 941 Sylvio X10 exclusive peptides (min: 557; max: 1249; std: 151). Most samples from Mexico displayed higher relative reactivity against peptides from the TcI/Sylvio genome (as well as several subjects from Colombia and the United States). Other subjects, particularly from Argentina, Brazil and Bolivia showed higher relative reactivity against TcVI/CL-Brener (but not exclusively). While the design of these arrays prevents a more detailed serotyping analysis, these serological reactivity signatures suggest differences in the infecting *T. cruzi* lineages in these subjects.

### Diversity of individual immune responses, revisited
In the discovery screening, we observed that most antigenic regions were private, followed by a long tail of shared antigens. Here, we revisited this analysis using data from the CHAGASTOPE-v2 arrays, where we focused on the 3868 clusters of antigenic regions identified at the discovery stage (3054 private/non-shared, 814 shared) now using 71 individual serum samples.

Each individual displayed reactivity to 5841 peptides on average, with a distribution that is in agreement with the observed large repertoire of private antigens. A similar distribution was also observed at the level of antigenic regions. Most of these antigenic regions represent novel antigens, meaning antigens whose sequence had no significant matches against those in the Immune Epitope Database (IEDB) (see Fig. 6A, B and Methods). Even for promising high-seroprevalence antigens, sequence similarity searches against the

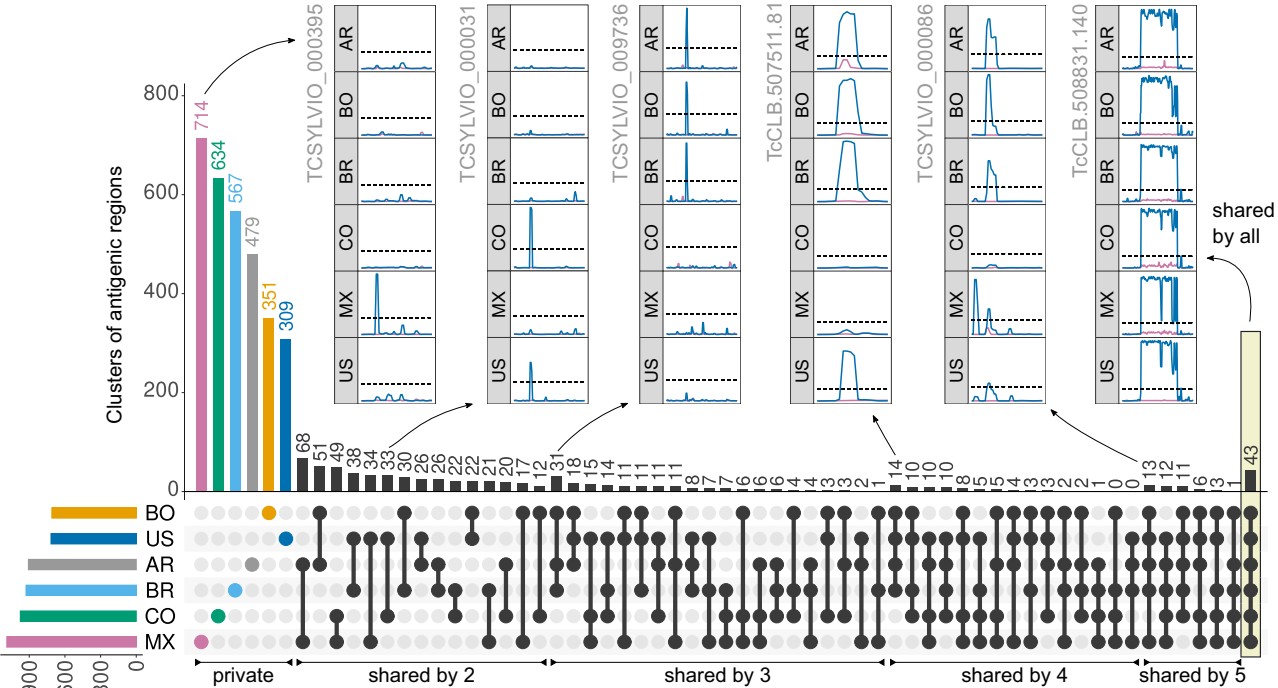

**Fig. 3 | Diversity of *T. cruzi* antibody-specific responses in pooled sera.** Non-redundant clusters of reactive regions in the analysed *T. cruzi* proteomes (clustered by sequence similarity) were counted on all intersections of the 6 analysed pooled samples. A cluster was antigenic in each sera pool when at least one of its regions was antigenic in that pool. The UpSet Plot[73] displays a histogram with these counts (top), as well as a visual depiction of all the set intersections (bottom, black). The coloured histogram at the bottom left shows the counts of total reactive clusters. Pooled samples are AR = Argentina; BR = Brazil; BO = Bolivia; CO = Colombia; MX = Mexico; US = United States. The insets show antibody-binding profiles (as in Fig. 2) for several selected antigens. Source data are provided as a Source Data file.

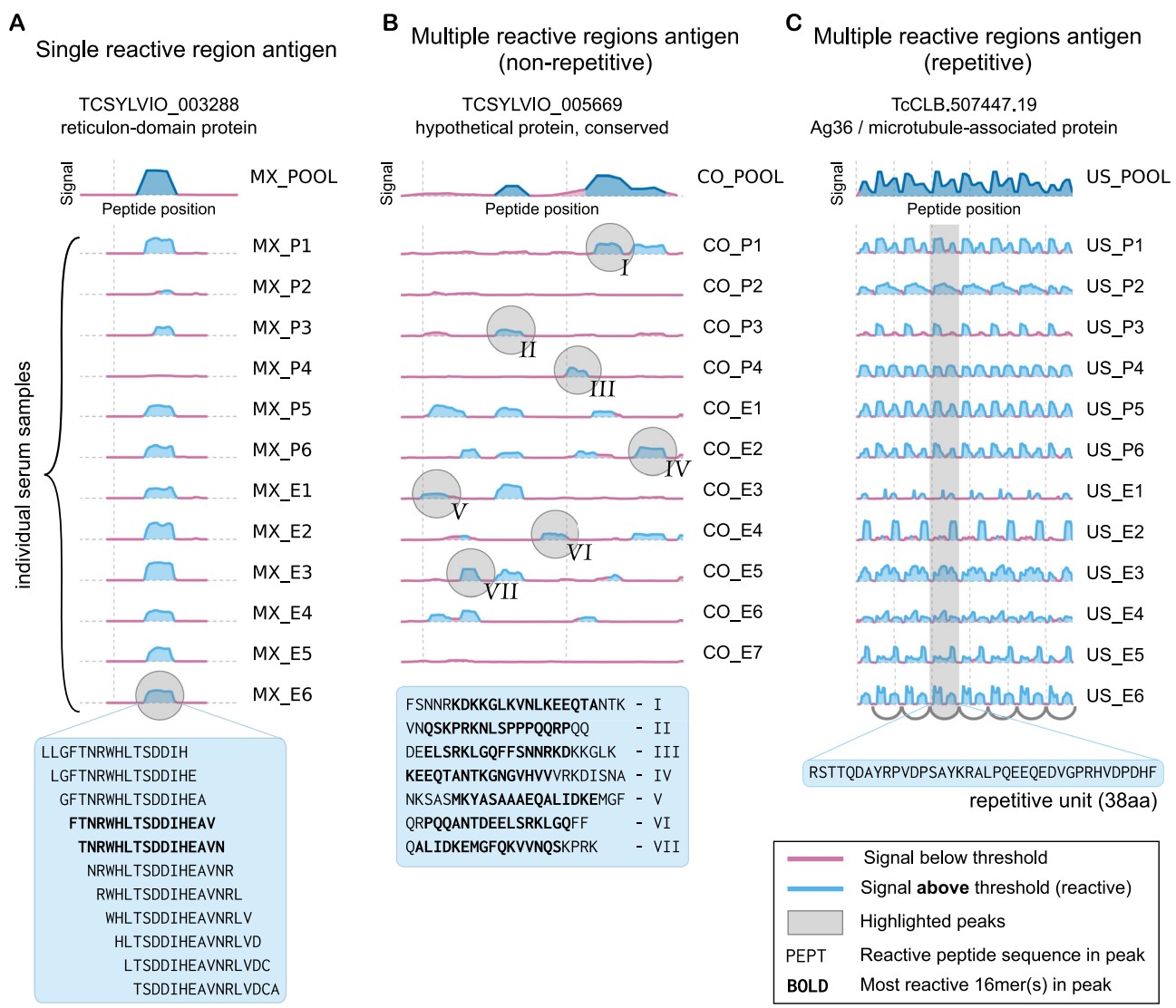

**Fig. 4 | Individual patient resolution and epitope mapping of Chagas disease antigens.** Examples of different types of antibody-binding profiles obtained using CHAGASTOPE-v2 arrays. In all cases, the reactivity of the pooled samples in the CHAGASTOPE-v1 discovery screening is shown at the top (dark blue), and stacked plots of the reactivity of the same protein against individual serum samples are shown below. See Supplementary Data S2 for the codes of patient serum samples (MX = Mexico, CO = Colombia, US = United States). Source data are provided as a

Source Data file. **A** Example antigen with a single reactive region, showcasing peptides with signal above threshold (and most reactive peptides in bold). **B** Non-repetitive antigen displaying multiple reactive peaks (marked using roman numerals), and the corresponding peak sequences below. **C** Repetitive antigen displaying heterogeneous recognition of the repetitive unit by different Chagas-positive individuals.

IEDB did not reveal significant matches for ~60% of these antigenic regions.

Interestingly, samples from southern South America shared more antigenic peptides with each other than samples from North America and northern South America (see Fig. 6C). As shown in the figure, with the exception of three samples (denoted with '*'), the two main branches in the heatmap with 28 and 33 samples map to these biogeographic divisions (see also Supplementary Fig. S4). A minor group of 10 samples of mixed origins showed reduced shared reactivities and clustered into a third branch.

Table 2 provides additional information on the clusters with higher seroprevalence in CHAGASTOPE-v2, highlighting 18 clusters detected by at least 70% of the subjects that did not show cross-reactivity against leishmaniasis-positive serum samples and did not match previously known antigenic epitopes (see Methods). The list of all 3,868 antigenic clusters and their seroprevalence in CHAGASTOPE-v2 can be found in Supplementary Data S7, while the full list of 9547

antigenic regions (redundant, unclustered) can be found in Supplementary File S5.

## Most human immune responses converge on the same fine epitope specificities

To investigate the modes of recognition of antigens and epitopes by different individuals, we performed alanine-scan mutagenesis of selected epitopes. Because substitution with alanine removes all side-chain atoms past the β-carbon, the effects of individual alanine mutations can be used to infer the roles of individual side chains. The CHAGASTOPE-v2 array design already included mutants designed in silico for 232 selected antigenic peaks found in the discovery screening. For each selected peak, the best 20mer sequence was used as a guide to probe antibody-binding by replacing each residue for an Alanine (except when the residue itself was Alanine, in which case we replaced it by Glycine, see Fig. 7A). This was done for all five overlapping peptides (16mers) in each antigenic sequence; hence, the same

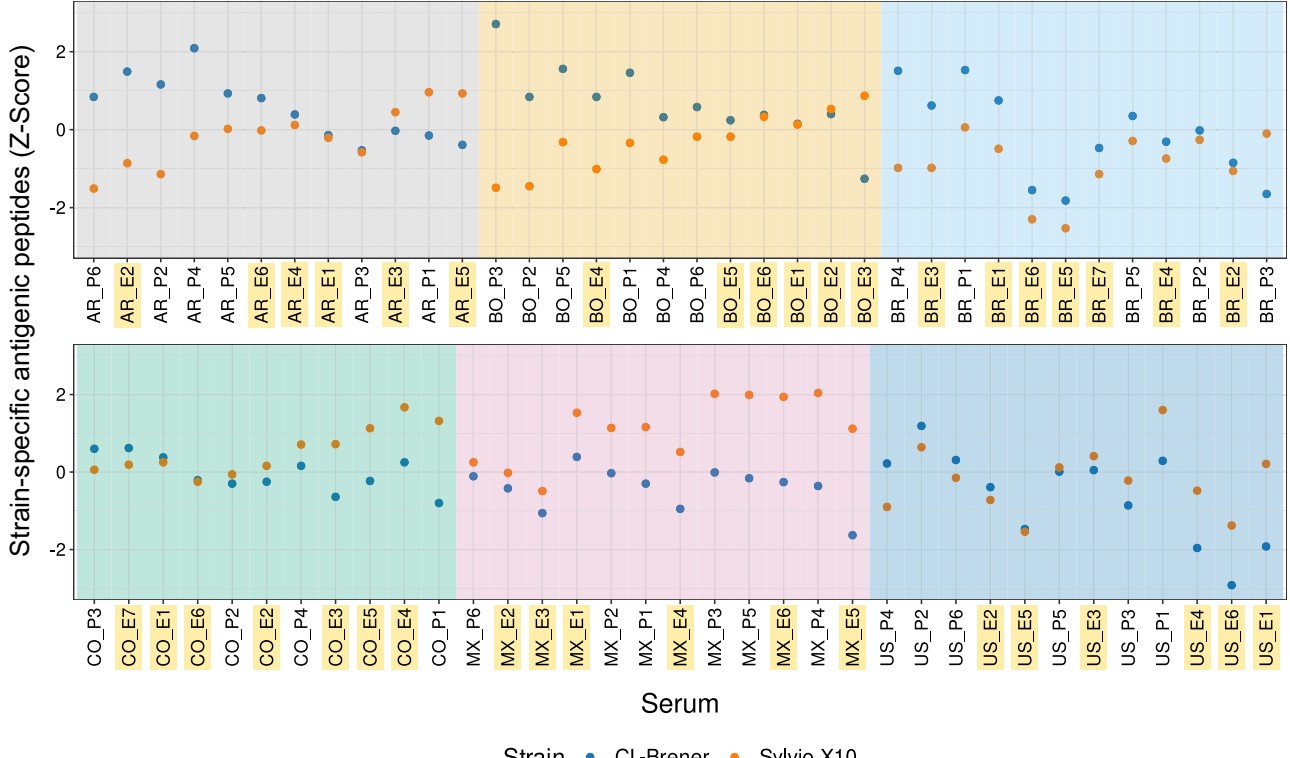

**Fig. 5 | Comparison of strain-specific antigenicity across subjects.** Counts of each subject's strain-specific reactive peptides (those present in only one strain and with a signal above the antigenicity threshold) were standardised using Z-scores. Standardisation was necessary because CL-Brener and Sylvio X10 strains have different numbers of encoded proteins (see Methods). Z-scores above 0 and below 0 represent the higher and lower relative numbers of strain-specific reactive peptides, respectively. See Supplementary Data S2 for the codes of patient serum samples (AR = Argentina, BR = Brazil, BO = Bolivia, CO = Colombia, MX = Mexico, US = United States). Samples that were not part of the discovery screening and hence were not used for antigen discovery and design of the CHAGASTOPE-v2 arrays are highlighted in yellow. Source data are provided as a Source Data file.

mutation was assayed several times, in different positions. Supplementary Fig. S5 and Methods provide a full description of the procedure.

When performing single-residue mutational scanning over different epitopes, we observed that antibody binding was diminished consistently for some amino acid residues. These key residues define the 'core functional' residues of each epitope, which represent residues likely in contact with the paratope in the Fab (antibody-binding) region of the immunoglobulins. As seen in Fig. 7 and Supplementary Fig. S6, the core residues were approximately 5–9 residues for these epitopes (9 was the most frequent value, with 65% of the epitopes having between 7 and 11 'core' residues, see Supplementary Fig. S7). Interestingly these core residues were mostly shared across positive sera (see Fig. 7), and hence, suggest that immune responses from different individuals converge on similar modes of binding.

As an example, the Ag2/CA-2 repetitive *T. cruzi* antigen[37], one of the most seroprevalent antigens in our screening, has a functional core defined by the AAGDK residues of the 12mer repetitive unit (see Fig. 7). Even individuals that showed low signal in the arrays or were borderline negative for this antigen with our signal threshold displayed the same residue fingerprint characteristic of this epitope (see Supplementary Fig. S8). This suggests that this convergent specificity on the antibody-binding mode is maintained across high- and low-titer responses.

Despite overall convergence on key residues, we also observed differences in the way that antibodies from some individuals bind to epitopes. For Ag2 the individual AR_E1 displayed higher relevance for residues 572Q and 573A (Fig. 7D) even when the signal profile of the protein was similar (Fig. 7E). Also, the high resolution of this technique allowed us to observe additional residues that in some individuals

produce unique effects on antibody-binding (when mutated). For example, at least seven individuals showed increased antibody-binding when 576G was mutated to an Alanine (with individual MX_E5 displaying the most extreme effect see Fig. 7C). Examples of other cases of differential epitope recognition by individual subjects are presented in Supplementary Figs. S9 and S10.

Similar observations were made for novel antigens (those without significant matches against the Immune Epitope Database, IEDB, see Methods). In the case of the Gim5A protein, the core epitope is defined by four core residues with a very high mutational signal (LPED) and three to four additional residues with more heterogeneity in the way that antibodies from different individuals recognise this epitope (see Supplementary Fig. S11). For example, the 136K-138A residues preceding the core epitope motif were relevant for antibody binding for about half of the positive serum samples for this antigen, whereas these residues were not relevant for the antibodies present in other subjects (e.g., BO_E6 and AR_E1).

The high resolution achieved by this strategy also allowed us to reveal cross-reactivity between epitopes that bear low sequence similarity. These antigens showed no detectable similarity to other antigens (hence they were not clustered together at the BLASTP stage) but displayed highly similar (or identical) key residues in this mutational analysis. One such example was a hypothetical protein (TCSYLVIO_002530) for which mutagenesis scanning revealed the characteristic AGDK core of the Ag2/CA-2 antigen (see Supplementary Fig. S6C and Supplementary File S6). There is almost no similarity between the protein sequence of Ag2 and this hypothetical protein, and even when looking at the scanned 20mer, there are only 5/6 shared residues. The fact that these residues constitute the core functional residues of the Ag2/CA-2 antigenic repeat suggests that, most likely, the Ag2 protein is

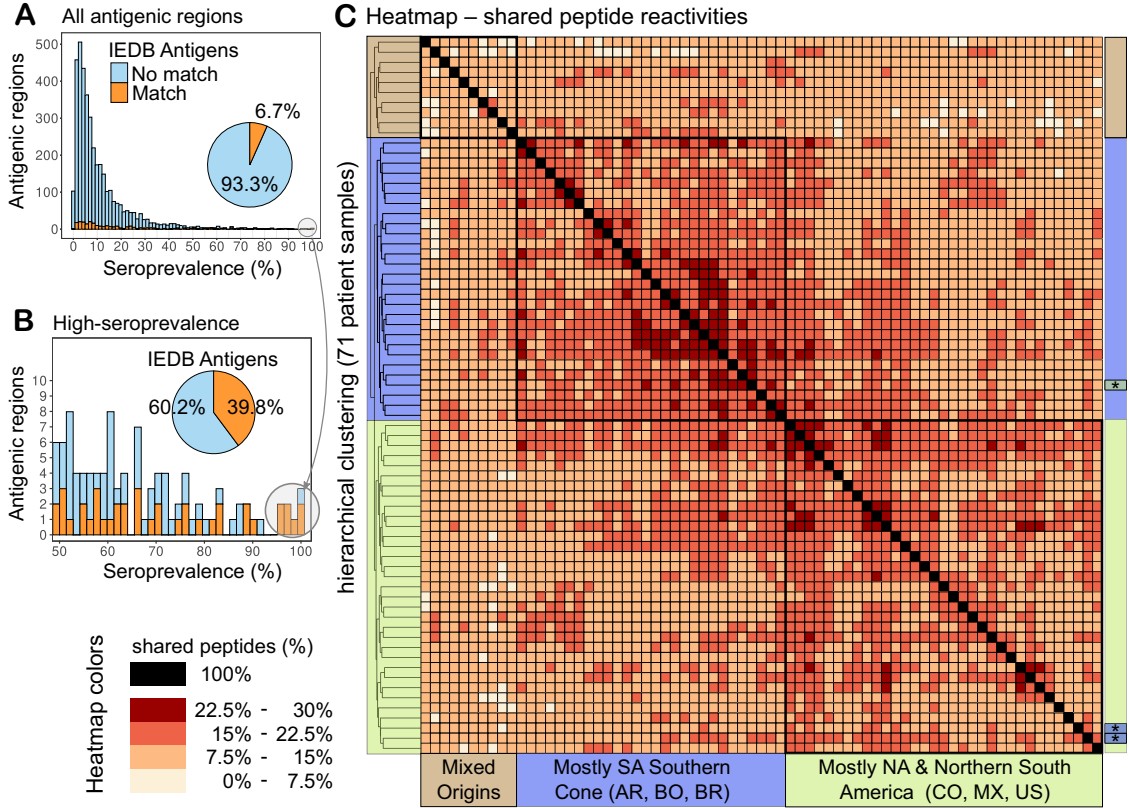

**Fig. 6 | Profiling of individual antibody responses reveals the diversity of anti-*T. cruzi*-specific antibodies. A** Histogram showing the number of clusters of antigenic regions that were reactive in each fraction of subjects (shown as seroprevalence). Sequence similarity to known antigens was assessed by BLASTP against *T. cruzi* linear epitopes in the IEDB (see Methods). Previously described antigens are shown in orange; antigens without matches to the IEDB are in light blue. **B** Inset zooming in on the 98 clusters reactive in at least 50% of the subjects (colours for novel and known antigens as in (**A**)). **C** Heatmap showing the percentage of non-redundant antigenic peptides that were shared between a pair of individual samples. The hierarchical clustering of samples shows three main branches (coloured and labelled at the sides) that correspond to the boxed peptide reactivities. A similar figure showing all sample labels is available, see Supplementary Fig. S4. Source data for each panel are provided as a Source Data file.

the natural immunogen while the TCSYLVIO_002530 protein contains a similar antigenic epitope in its sequence.

Examples of other antigenic epitope cores embedded in otherwise non-conserved sequences are shown in Supplementary Fig. S6C: the beta-adaptin 3 protein (TCSYLVIO_008280) with a core functional epitope highly similar to that of the trans-sialidase protein (TcCLB.459061.10), and the nucleolar RNA-binding protein (TcCLB.510859.17) with a functional core similar to the Rieske iron-sulfur protein (TcCLB.510759.120), amongst others (see also Supplementary File S6). In all these cases, the normalised signal observed in the arrays for the true positive antigens was ~2-fold higher than the signal observed for the candidate cross-reactive proteins.

A detailed account of all the core epitope motifs revealed by this study will likely be further explored by researchers in the field. Supplementary Data S8 and S9 provide a summary of the key residues in other epitopes analysed. Similar visualisations of these data are provided for all mutagenised epitopes in Supplementary File S6. A separate analysis focused on known repetitive antigens is provided in Supplementary Fig. S12 and Supplementary File S7.

### Novel antigens for improved diagnosis of Chagas disease

We next explored the impact of novel antigens and their identified linear epitopes on the diagnosis of Chagas disease using an independent technology, fluorescent-linked immunosorbent assays (FLISA). For this, we focused on challenging samples for diagnosis such as serodiscordant samples and those from specific human populations where current commercial tests show suboptimal

performance (e.g. Mexico)[20,38]. For the selection of antigens, we prioritised those identified herein without matches in the IEDB, and with high signal and either high global seroprevalence or high seroprevalence when restricted to samples from North American origins (CHAGASTOPE-v2 array data). The list of selected antigens includes TCSYLVIO_004530 (CAR-Ag1, top hit in Table 2; 99% global seroprevalence); TCSYLVIO_005782 (CAR-Ag2, Gim5A protein, 66% global seroprevalence); TcCLB.507195.60 (mucin-associated protein, CAR-Ag5, 49% global seroprevalence which increases to ~74% when restricted to MX + US samples); TcCLB.511029.20 (Kinetoplast-associated protein 3, CAR-Ag9, 66% global seroprevalence, increasing to ~74% when considering only MX + US samples); and TcCLB.511593.60 (mucin-associated protein, CAR-Ag12 with ~50% global seroprevalence but ~74% seroprevalence when considering only the reactivity observed in MX and US samples); and TCSYLVIO_005669 (hypothetical protein, CAR-Ag24, see Fig. 4, 86% global seroprevalence). Two of these antigens had several reactive peaks spanning a relatively long antigenic region; hence, we produced these antigens as recombinant proteins (CAR-Ag1 and CAR-Ag24). The other antigens were produced as short synthetic peptides (see Methods).

Using a panel of 17 serodiscordant samples from Argentina, we investigated the performance of these identified antigens for the detection of confirmed cases (see Methods and Supplementary Data S10). In independent FLISA assays, three of these antigens were able to detect three (CAR-Ag12; Sn = 25%; Sp = 100%); five (CAR-Ag24; Sn = 42%; Sp = 100%); and six (CAR-Ag1; Sn = 50%, Sp = 100%) of 12

**Table 2 | Selection of antigens reactive in most Chagas-positive individual serum samples**

| Cluster | | Best antigenic region in cluster | | | | | |
|---|---|---|---|---|---|---|---|
| # Regions | Sero (%) | Locus identifier | Start—End | Description | Sero (%) | Rep | Best 16mer |
| 169 | 100 | TCSYLVIO_004530 | 9–244 | Trans-sialidase, putative | 99 | No | GTNSDPDSFSSTNVSG |
| 3 | 92 | TcCLB.509007.70 | 1249–1308 | Protein phosphatase 2C | 90 | No | AVPNFAAATADKPVGT |
| 48 | 87 | TcCLB.506289.240 | 161–312 | Hypothetical protein | 55 | No | EGKDERKSGEATAPQV |
| 8 | 87 | TcCLB.507297.30 | 1–96 | Metacaspase | 86 | No | ERPPRVDVEEFFQQAE |
| 10 | 83 | TcCLB.506597.33 | 145–193 | Hypothetical protein | 80 | No | LRQIDASPPEPFTAAP |
| 3 | 80 | TCSYLVIO_004283 | 221–276 | 60S ribosomal protein L4 | 80 | No | KEAMAFLKAIGAVDDV |
| 2 | 79 | TCSYLVIO_009331 | 481–536 | SMC protein 4 | 79 | No | LRKIDAATERNGNLVA |
| 1 | 77 | TcCLB.506941.194 | 1025–1080 | Hypothetical protein, conserved | 77 | No | FRKIDAAVPVPNTSYA |
| 1 | 76 | TcCLB.507809.119 | 81–140 | Autophagy protein Apg6 | 76 | No | VLRKIEEEFQQLEEQK |
| 2 | 76 | TCSYLVIO_009815 | 281–336 | Hypothetical protein | 76 | No | FRQIDTEHNDRITAEQ |
| 2 | 75 | TCSYLVIO_001410 | 229–284 | NADH-cytochrome b5 reductase | 75 | No | PPFMEAISGDKDFKTS |
| 2 | 73 | TCSYLVIO_003288 | 97–152 | Reticulon domain protein | 73 | No | LTSDDIHEAVNRLVDC |
| 3 | 72 | TcCLB.506441.20 | 641–888 | Cytoskeleton-associated protein | 70 | Yes | REGRERGYPEEKEDSR |
| 1 | 72 | TcCLB.506925.460 | 41–92 | Kinetoplastid kinetochore protein 16 | 72 | No | LRMIDELAAGVEMWKQ |
| 3 | 72 | TCSYLVIO_003590 | 89–140 | Microtubule-associated protein Gb4 | 72 | No | LREIDDVENHASQSRA |
| 23 | 72 | TcCLB.510157.10 | 281–332 | MASP subgroup S022 | 45 | No | AASANKYDTVPQSAGS |
| 3 | 70 | TcCLB.509139.20 | 37–88 | Hypothetical protein, conserved | 70 | No | SGGAPPTRGGFGAGTS |
| 2 | 70 | TcCLB.509791.120 | 65–172 | Kinetoplast-associated protein | 70 | No | YDVSKPLDVEKEISKA |

A list of the 18 clusters of antigenic regions, which showed reactivity against at least 70% of the positive individual serum samples, did not show cross-reactivity against leishmaniasis-positive serum samples in the previous assay, and for which at least 90% of its regions were not similar to already known antigenic epitopes found in the IEDB. For each cluster, we show a representative antigenic region with the highest seroprevalence in CHAGASTOPE-v2. All 3868 antigenic clusters can be found in Supplementary Data S7.
*# Regions* number of regions in said cluster, *Sero* seroprevalence, *Rep* repetitive (contains internal tandem repeats).

positive samples with serodiscordant tests at origin (see Fig. 8A). In concert these three identified antigens were able to rescue six (50%) of these serodiscordant samples with nil reactivity against negative samples from the same collection.

These antigens were also investigated against a panel of 168 samples from two regions in Mexico (Oaxaca, Yucatán, see Supplementary Data S10). As shown in Fig. 8B, these antigens were specific for *T. cruzi*, with varying degrees of sensitivity. In both panels, the best-performing antigen was CAR-Ag1, a fragment of a non-repetitive member of the trans-sialidase superfamily. Performance of CAR-Ag1 in these two panels of samples was compared against the Wiener Chagatest serological reagent, which is an FDA-cleared test[38]. As summarised in Fig. 8C, the identified CAR-Ag1 antigen alone outperformed the combined reactivity of the six *T. cruzi* antigens included in the Wiener v4 assay[39]. Besides validating the peptide microarray platform for the discovery and mapping of antigens, this work shows the impact of discovered antigens on the improvement of Chagas disease diagnostics in challenging cases.

## Discussion

We have produced a detailed map of the antibody specificities against linear epitopes in patients with Chagas disease, described in depth here for the first time. Previous studies show that some pathogens induce short-lived, polyclonal plasma cells to dilute long-lived, specific antibody responses[40]. In contrast, *Trypanosoma cruzi* induces a massive clonal expansion of B-cells during the acute phase leading to the production of parasite-specific and autoreactive antibodies as well as high levels of antibodies that were described as having unknown specificity[41,42]. Our description of a large and diverse number of specificities, composed of mostly non-shared epitopes (low seroprevalence at the population level), supports these previous observations and provides information on the targets of these antibodies.

The power of these massively parallel serological assays lies in the delineation of the responses of individual patients to each identified epitope, hence producing a rich human seroprevalence matrix for these antigens. The analysis of non-redundant clusters of antigenic

regions revealed that most of these are novel antigens, including those that were detected by at least 50% of the individuals (59 out of 98 of these did not have matches against the IEDB), and are thus of interest for development and optimisation of immunoassays.

Interestingly, when analysing shared peptide reactivities across individual samples, we observed that while shared reactivities were low overall (<30%, see Fig. 6), they displayed a pattern that matches the current knowledge of American biogeographic regions[43,44] as well as the serological separation of the *T. cruzi* population into two major lineages[45]. It is thus tempting to speculate that these antibody repertoires are shaped in concert by both host and parasite genetics. Analysis of the reactivity of serum samples against strain-specific peptides from either the CL-Brener or Sylvio X10 strains also supports this pattern, with a greater strain-specific reactivity against CL-Brener (TcVI evolutionary lineage) in samples from the southern cone of South America; and a relatively greater strain-specific reactivity against Sylvio X10 (TcI lineage) in samples from North America (particularly Mexico) and northern South America (Colombia).

We validated a number of identified antigens in an independent multiwell immunosorbent assay platform with improved diagnostic performance in two challenging panels. In a panel of serodiscordant samples and in a panel where current tests show suboptimal performance, these antigens were able to rescue a number of samples that otherwise would have been undiagnosed. Further exploration of combinations and optimisation of these antigens is underway. Two of the identified antigens assayed (CAR-Ag5, CAR-Ag12) are mucin-associated proteins (MASPs). MASPs represent a large family of highly diverse antigenic proteins[46,47]. Previously identified MASP epitopes that were antigenic in humans[47] did not show high relative seroprevalence in our study. This is in agreement with current evidence: not all MASP genes are expressed simultaneously[48]; antibodies against one MASP family member may only detect a few parasites from a population[46,49]. All this evidence supports the low seroprevalence observed for most MASPs. However, interestingly, the high seroprevalence of CAR-Ag5 and CAR-Ag12 antigens in diverse samples across the Americas suggests that a few MASP

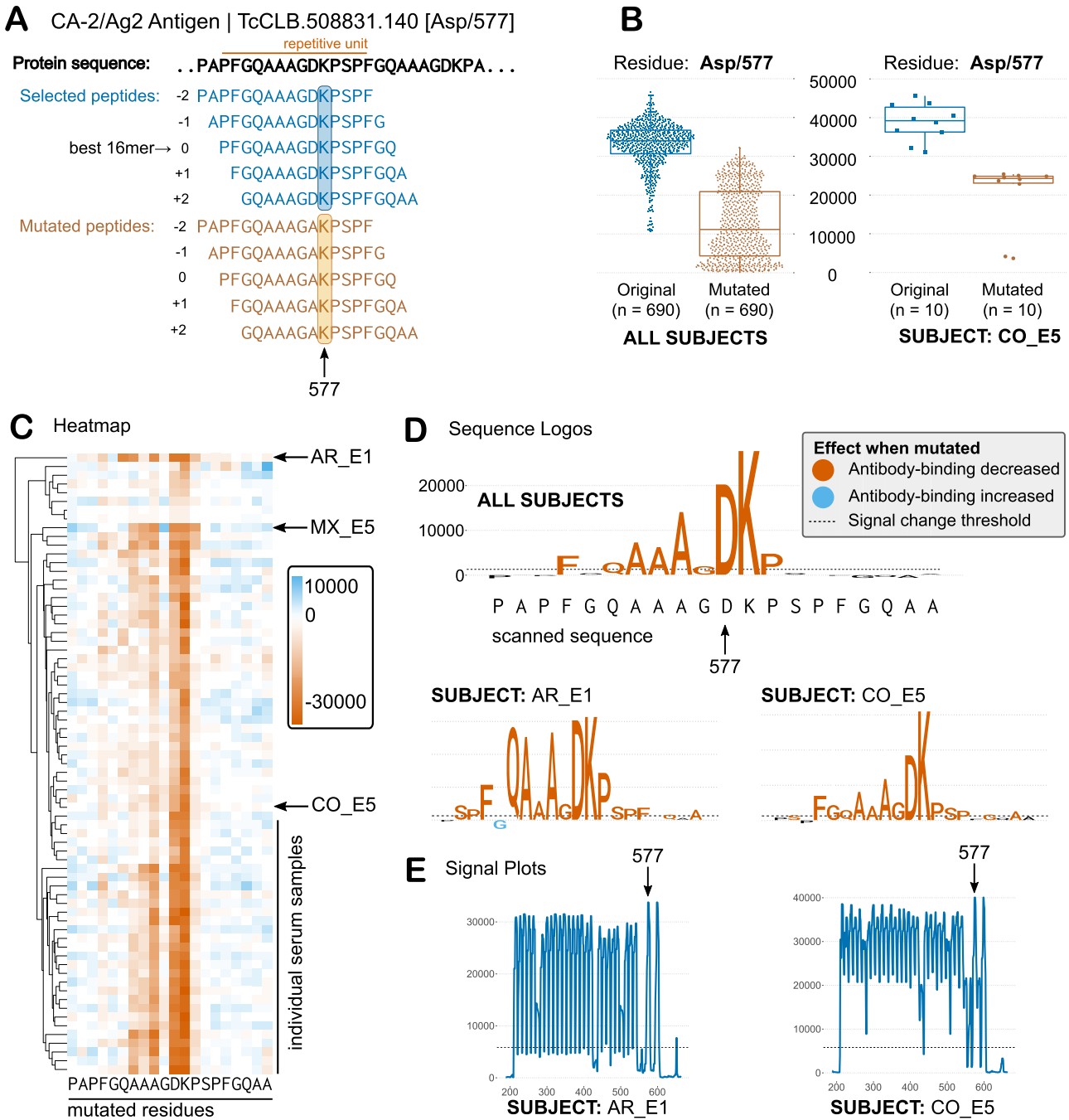

**Fig. 7 | Single-residue scanning mutagenesis of *T. cruzi* antigens. A** Schematic representations of single-residue mutagenesis for one repetitive antigen (only 1 residue is mutated in this depiction for clarity). **B** Average signal of the original and mutated peptides for the Asp/577 residue for all positive sera (*n* = 69 biological independent samples) for this antigen (left) and for one selected subject (right). Each point represents the signal from an independent experiment, e.g., a peptide that contains residue 577 either as Asp (wild-type) or Ala (mutated). Boxplots: the upper and lower bounds of the box correspond to the first and third quartiles. Whiskers extend from the box up to 1.5 × IQR (interquartile range) or to the smallest and/or largest value. The centre of the box corresponds to the median value. **C** Heatmap showing the effect of mutations on antibody binding (signal change from original to mutated peptides) for all residues and for all sera. Mutations that decrease antibody binding are shown in different shades of orange, while those that increase binding are shown in shades of blue. Columns = residue positions, Rows = individual serum samples. **D** Sequence logos summarise data for all positive sera (Core residues), or for individual cases (*y*-axis: signal change in fluorescence units). Colours follow the heatmap. **E** Antibody-binding signal plots for selected subjects. Source data for each panel are provided as a Source Data file.

members may be conserved and expressed in human infections by different *T. cruzi* strains.

A consequence of the experimental platform used to detect antibody-binding is the inherent bias towards short linear epitopes, likely missing most conformational epitopes. This is evident in our failure to detect antibody binding to at least one known bona fide antigen: Ag1[36], also known as FRA or JL7[50], a cysteine protease (clan CA,

family C2, CL-Brener Locus ID: TcCLB.505985.9) which is a component of commercial kits for the diagnosis of *T. cruzi* infection. No reactivity was observed in our short peptide screenings against the reported antigenic sequence, suggesting that the epitope(s) in this antigen may be conformational.

We also identified and reported 888 cross-reactive peptides against American Tegumentary Leishmaniasis (data in Supplementary

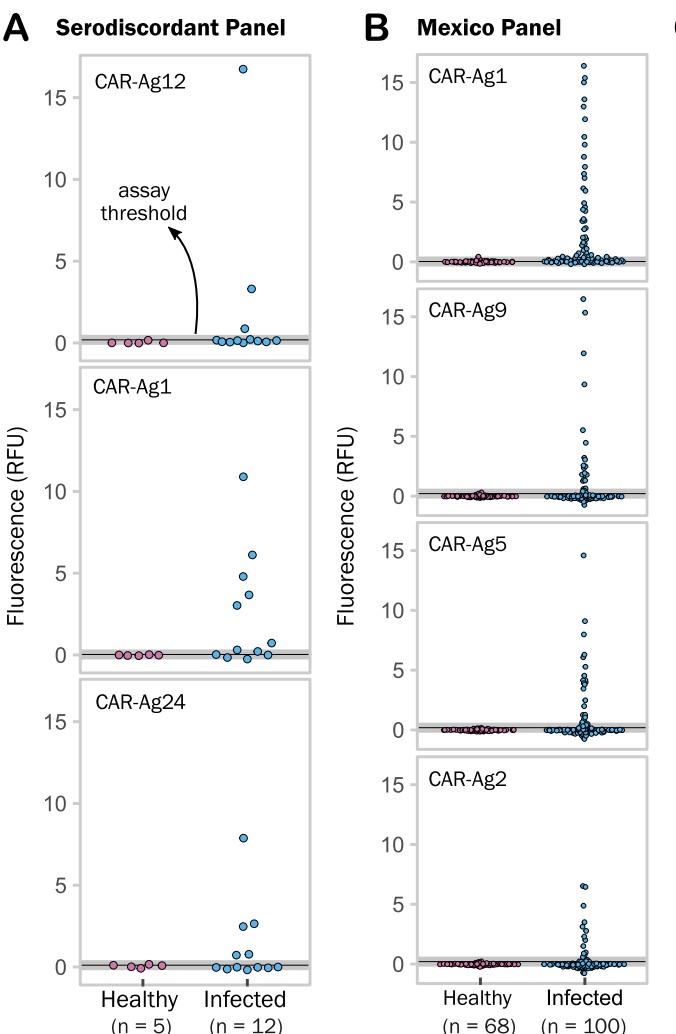

**C** **Summary of Performance (CAR-Ag1)**

| Serodiscordant panel (n = 17) | | |
|---|---|---|
| | **Wiener v4** | **CAR-Ag1** |
| AUC | 0.74 | 0.80 |
| AUC (95% CI) | 0.49 – 0.98 | 0.57 – 1.00 |
| Sensitivity | 0.50 | 0.75 |
| Specificity | 1.00 | 1.00 |
| PPV | 1.00 | 1.00 |
| NPV | 0.45 | 0.62 |
| Cohen's kappa | 0.65 | |
| Kappa (95% CI) | 0.32 – 0.99 | |

| Mexico panel (n = 168) | | |
|---|---|---|
| | **Wiener v4** | **CAR-Ag1** |
| AUC | 0.69 | 0.82 |
| AUC (95% CI) | 0.61 – 0.77 | 0.75 – 0.88 |
| Sensitivity | 0.44 | 0.60 |
| Specificity | 1.00 | 0.95 |
| PPV | 1.00 | 0.95 |
| NPV | 0.53 | 0.61 |
| Cohen's kappa | 0.66 | |
| Kappa (95% CI) | 0.55 – 0.78 | |

**Fig. 8 | Performance of identified antigens for diagnosis of challenging samples.** All assays are fluorescent-linked immunosorbent assays (FLISA), each dot in a plot represents the average signal of duplicates from an individual sample. RFU means relative fluorescence units. Assay threshold is shown as a black line and corresponds to the value of the mean signal of all healthy individuals plus three standard deviations. An indeterminate zone around the threshold is shown in grey. **A** Reactivity of three antigens against a panel of serodiscordant individuals (see Methods); in this case, the classification of Healthy vs Infected is based on the result of the confirmatory technique (IIF indirect immunofluorescence, performed with replicates in a reference centre by trained personnel). **B** Reactivity of four antigens against a panel of samples from Mexico. Individuals were classified into Healthy vs Infected using both serology and molecular tests at the origin. **C** Performance of the CAR-Ag1 antigen against the Wiener Chagatest recombinant v4 assay. Cohen's kappa index measures concordance between the two assays in each panel. Source data for each panel are provided as a Source Data file.

Data S6), a co-endemic disease caused by a related trypanosomatid parasite. This is important, as many Chagas disease false positives are suspected to be a consequence of *Leishmania spp.* cross-reactivity[51]. While the number of leishmaniasis samples in the pool may be small (*n* = 6), the ability to screen at this scale outweighs this potential limitation. Cross-reactivities to other parasites, bacteria or viruses have not been studied and will have to be explored when validating these epitopes.

Extensive mutational scanning of a large epitope set revealed key residues for antibody binding in each case. The precision of this type of analysis led to the identification of cases where antibodies from different subjects converged to a shared mode of antibody recognition for the studied epitope, and cases where the same epitope had divergent (alternative) modes of recognition by different individuals (see Fig. 7, Supplementary Figs. S8 and S9, and Supplementary File S6). Both convergence and divergence in B-cell clonal lineages have been observed[52] leading to recurring motifs for the recognition of viral protein antigens[53]. These mutagenesis data also represent a rich resource to guide the optimisation of immunoassay reagents.

To our knowledge, this Atlas is the largest collection of Chagas disease antigens and epitopes described to date, and the first dataset providing fine resolution of seroprevalence to epitopes in humans. Due to the breadth and diversity of the clinical samples analysed, this study also provides a large set of experimentally validated negative data (non-antigenic proteins and peptides). This is almost always overlooked, but it represents a highly valuable dataset for the training of predictors, which often needs to work under the assumption that proteins with no previous information on their antigenicity are non-antigenic[54,55]. The datasets from the primary discovery screening also provide a large corpus of data on dominant *T. cruzi* peptides reactive to sera from healthy subjects from different human populations.

The data produced in this study reflect a snapshot in time of the antibody repertoires of each subject. Many questions about these repertoires remain. What is the nature of the private (non-shared) set of antibody specificities? Which epitopes are the targets of short-lived responses? And which are the targets of long-lived responses? The observed low seroprevalence of a large fraction of antigens may be explained if this is a fluctuating repertoire. It is thus tempting to

speculate that private antigens (as described in this work) may be the target of short-lived or weak antibody responses. Under this scenario, the B-cell clones producing these antibodies may decay after some time, and thus the observed feature of being unique to one or very few individuals in these snapshots may be the telltale of these short-lived immune responses. This agrees with the current view of the complex and focal dynamics of Chagas disease in the host[56–58], where waves of parasite burst from different foci at different moments may direct the immune response to antigens that are uniquely expressed or exposed in different tissue environments.

In chronic Chagas disease, antibody levels are maintained by the persistence of parasites and antigens[59]. Peptidic biomarkers in this Atlas provide essential information to study the dynamics of the more prevalent (shared) antibody specificities and how they fluctuate upon reduction of parasite loads or elimination, which would provide much-needed markers for novel patient follow-up in clinical trials.

The Human Chagas Antigen and Epitope Atlas is a reference resource that is freely accessible. The resource generated and described herein comprises the collection of antigenic regions of the *Trypanosoma cruzi* pangenome, as revealed by analysing the anti-*T. cruzi* human antibody repertoires of 71 Chagas disease patients. These individual antibody repertoires described in detail represent a foundational resource for the community that will serve as a major accelerator for the development of new diagnostics, serology-based immunoassays vaccines, and to study of the dynamics of adaptive immune responses at high resolution.

## Methods

All procedures followed the Declaration of Helsinki Principles. Written informed consent was obtained from all individuals (or from their legal representatives), and all samples were decoded and de-identified before they were provided for research purposes. This study was approved by the Institutional Review Board of the Instituto de Investigaciones Biotecnológicas, Universidad Nacional de San Martín, and by the ethics committees of each clinical site (see below). No compensation was provided to participants.

### Human serum samples

Human serum samples from *T. cruzi*-infected patients and matched negative subjects used in this study were part of the collections of the Laboratorio de Enfermedad de Chagas, Hospital de Niños 'Dr. Ricardo Gutierrez' (HNRG, Buenos Aires, Argentina) (AR; $n = 18$); Fundación CEADES (Cochabamba, Bolivia) (BO; $n = 17$); Protozoology Laboratory (LIM 49), Hospital das Clínicas, Faculdade de Medicina, Universidade de São Paulo (São Paulo, Brazil) (BR; $n = 18$); Instituto Nacional de Salud Pública (Tapachula, Mexico) (MX; $n = 18$); Instituto de Cardiología (Bucaramanga, Colombia) (CO; $n = 15$); University of South Carolina (South Carolina, USA) (US; $n = 18$). Human serum samples from patients with American Tegumentary Leishmaniasis (ATL) and matched negative subjects were from the Instituto de Patología Experimental, Universidad Nacional de Salta (IPE, Salta, Argentina) (LE; $n = 12$). A list of samples and their code identifiers as used in this work are provided in Supplementary Data S2. Chagas disease patients were in the asymptomatic chronic stage of the disease without cardiac or gastrointestinal compromise (age range: 15–96 years, median: 48 years). Serum samples were collected from clotted blood obtained by venipuncture and analysed for *T. cruzi*-specific antibodies with the following commercially or in-house kits: AR: Chagatest ELISA lysate (Laboratorios Wiener, Argentina), Chagatest HAI (Laboratorios Wiener, Argentina), Chagatest ELISA recombinant v4.0 (Wiener Lab, Argentina); BO: recombinant and lysate ELISA; BR: conventional in-house ELISA (confirmed by TESA-Blot[60]); CO: ELISA (Chagatek, Organon, Argentina) and HAI (Chagatest, Weiner, Chile); MX: Bio-Rad Chagascreen Plus v4 (recombinant), Test ELISA para Chagas III (Grupo Bios, Chile), Accutrak Chagas Microelisa Test System (Laboratorio

Lemos, Argentina), Accutrack Chagatek ELISA recombinante (Laboratorio Lemos, Argentina); US: Chagas Stat-Pak (Chembio, Medford, NY), Hemagen Chagas' Kit ELISA (Hemagen Diagnostics Inc., Columbia, MD), and an in-house TESA-Blot[60]. ATL samples were classified using an in-house ELISA based on crude antigen extracts from promastigotes and amastigotes of three different endemic *Leishmania* species and two reference strains[61]. Two additional panels of samples were used for the validation of antigens and assessment of diagnostic potential. One set of discordant samples (ARSD, $n = 17$) was obtained from patients that were previously examined by indirect hemagglutination (IHA, Chagatest HAI, Wiener Lab, Argentina) and ELISA (Chagatest ELISA recombinant 4.0, Wiener Lab, Argentina) and that exhibited discordant results even after both methods were repeated at least twice. The serological status of these patients was defined by indirect immunofluorescence (IIF) performed in a reference centre by trained personnel (Central Laboratory of the Santa Fe Province). In all serum samples included, the result given by the third reaction was in accordance with clinical and epidemiological information of the patient obtained by a physician at the same centre[62]. A second set of samples (MXO, MXY, $n = 168$) was obtained from patients in Mexico (MXO = Oaxaca; MXY = Yucatan) and were classified into *T. cruzi*-infected or negative (healthy) based on four distinct serological tests (see above). The procedures were approved by the following ethics committees: Hospital de Niños 'Ricardo Gutierrez' (#CEI 14.14); Fundación CEADES (CE-CEADES-4-12-2018; IRB 0990-0279; FWA: 00024189); Comité de Etica en Investigación, Instituto Nacional de Salud Pública, Mexico (CI: 1369, Registro ante CONBIOÉTICA: 17CEI00420160708, Registro ante COFEPRIS: 13 CEI 17 007 36, FWA: 00015605), Comité de Ética en Investigación FOSCAL, Bucaramanga, Colombia (CEI 21/11/14) and Fundación Cardioinfantil (Acta 512/2015), the study sites of the CHICAMOCHA3-equity trial; Baylor College of Medicine (Houston, TX, USA) (#H-35471 and #H-32321), the Gulf Coast Regional Blood Center (Houston) (#13-002), and the South Texas Tissue and Blood Center (San Antonio, TX, USA); the Comisión Provincial de Investigaciones Biomédicas, Ministerio de Salud Pública, Gobierno de la Provincia de Salta, Argentina (Expte 321 – 136934/2018); and the Ethics Review Board of the Santa Fe Province (RP N° 300). Samples from LIM 49 were part of an older collection of samples (8–10 years) and qualify as secondary research use of biospecimens for which informed consent was not required. These fall into exemption #4 in the list of exemptions for the requirement of informed consent developed by the Office for Human Research Protections (OHRP), US Department of Health & Human Services, as it did not involve new recruitment of human participants and samples did not include any direct identifier.

### Array designs

**CHAGASTOPE-v1: design used for antigen and epitope discovery.**
Two *T. cruzi* proteomes were used in this design: Sylvio X10[63] and CL-Brener[31], both retrieved from TriTrypDB Release 5 (2016)[64]. To produce a tiling display of peptides, we first merged sequences from both proteomes and parsed all proteins, removing those shorter than 16 amino acids and duplicates (based on protein ID). This resulted in 30,500 proteins: 10,832 for Sylvio X10 and 19,668 for CL-Brener.

Because we needed to display complete proteomes in the form of tiling peptides covering all proteins, it was necessary to optimise the use of the peptide array capacity. In a previous study for 457 *T. cruzi* proteins[12] we fine-mapped epitopes using overlapped peptides with a sliding window of one amino acid residue (maximum resolution). Here we performed simulations on those data where we increased the offset between overlapping peptides while measuring the success at mapping the epitopes of known antigens (see Supplementary Fig. S13). Performance at this task was measured by the Area Under the ROC Curve. Performance was high and stable for offsets 1–3 (AUCs 0.825 to 0.96) and showed only a slight decrease with an offset of 4 (AUC 0.76 to 0.93). Greater spacing between peptides (less overlap) affected

performance down to an AUC of 0.66 to 0.79 for an offset of 10. Based on this information, the number of peptides in the 30,500 proteins and on the space available on our microarray, we split proteins into peptides of length 16 with an offset of 4 amino acids between consecutive peptides, meaning that there was an overlap of 12 amino acids between those peptides.

We then removed duplicate peptide sequences; thus, each peptide was placed only once in the microarray. This set of 2,441,908 unique peptides can be found in Supplementary File S8 and in the submission to the ArrayExpress Database. Finally, we added other peptides from a number of sources to the CHAGASTOPE-v1 design, such as positive controls that corresponded to previously identified antigenic regions[12], and peptides from other pathogens that are seroprevalent in humans (e.g., cytomegalovirus). These positive controls were repeated four times across the array as peptides of length 15 with an offset of 1 amino acid to have a higher resolution for epitope mapping and to match the original conditions of past works. This, along with the trimming of a few peptides in the synthesis (see 'Array synthesis' below), resulted in a final array design containing 2,842,420 peptides, all of which were present only once in the microarray except for the positive controls. These peptides were assigned randomly to spots in the microarray.

**CHAGASTOPE-v2: design used for characterisation of antigenic regions.** Because the aim of this second design was to analyse a smaller subset of peptides in higher detail, we focused on the 9547 antigenic regions found in the first design. We produced tiling displays of peptides spanning these regions, using peptides of length 16 with an offset of 1 amino acid (maximal resolution for epitope mapping), which resulted in 242,154 unique peptides. These peptides can be found in Supplementary File S9 and in the submission to the ArrayExpress Database. In this array design, we also included additional peptide variants from the *T. cruzi* 231 strain (DTU TcIII)[65] that mapped to these regions as well as a number of detailed mutagenesis scans (AlaScan) of selected epitopes (see main text). The final array design contained 392,299 addressable peptide spots. Peptides were assigned randomly to these spots. This design was used to drive the synthesis of QX12 (12-plex) arrays where the same CHAGASTOPE-v2 design was replicated across all 12 sectors of the array (assayed individually).

**Array synthesis**
The CHAGASTOPE array designs were synthesised at Roche Sequencing Solutions (Peptide Lab, Madison WI, now Nimble Therapeutics) with a Roche Sequencing Solutions Maskless Array Synthesizer (MAS) by light-directed solid-phase peptide synthesis using amino-functionalised support (Greiner BioOne) coupled with a 6-aminohexanoic acid linker and amino acid derivatives carrying a photosensitive 2-(2-nitrophenyl) propyloxycarbonyl (NPPOC) protection group (Orgentis Chemicals). Amino acids (final concentration 20 mM) were pre-mixed for 10 min in N,N-Dimethylformamide (DMF, Sigma Aldrich) with N,N,N',N'-Tetramethyl-O-(1H-benzotriazol-1-yl) uronium-hexafluorophosphate (HBTU, Protein Technologies, Inc.; final concentration 20 mM) as an activator, 6-Chloro-1-hydroxybenzotriazole (6-Cl-HOBt, Protein Technologies, Inc.; final concentration 20 mM) to suppress racemization, and N,N-Diisopropylethylamine (DIPEA, Sigma Aldrich; final concentration 31 mM) as a base. Activated amino acids were then coupled to the array surface for 3 min. Following each coupling step, the microarray was washed with N-methyl-2-pyrrolidone (NMP, VWR International), and site-specific cleavage of the NPPOC protection group was accomplished by irradiation of an image created by a Digital Micro-Mirror Device (Texas Instruments), projecting 365 nm wavelength light. Coupling cycles were repeated to synthesise the full in silico-generated peptide library. Coupling cycles were limited to avoid long synthesis times, which had the consequence of trimming some peptides in our design by a few

amino acids (usually peptides where a single amino acid appeared many times). This occurred in 0.5% of the peptides in the first design and 1.4% of the peptides in the second one, with an average of 1.5 and 1.7 amino acids trimmed in each case, respectively. Because this was a rare event, and also because the trimming removed only one or two amino acids, and further because we also smoothed the signal data using a rolling median technique (see below), we assumed this trimming had no substantial impact on the analysis of the data. To check that the length of the peptides had no negative impact on their synthesis, we analysed how the average raw antigenicity signal and its standard deviation vary according to the position of the epitope within assayed 16mer peptides. We observed that the measured antibody-binding was strong as long as the epitope was present within the peptide (see Supplementary Fig. S14).

**Array assays**
For the antigen and epitope discovery screening using the CHAGASTOPE-v1 design, we produced and assayed 28 1-plex high-density peptide arrays. For the epitope characterisation, mapping and seroprevalence study using the CHAGASTOPE-v2 design, we produced and assayed 12 12-plex sectorised high-density peptide arrays. Supplementary Data S1 provides an overview of 1-plex and 12-plex arrays used in this work, while Supplementary Data S3 provides a list of arrays slides and samples used in each assay.

**Sample binding and detection**
Prior to sample binding, the final removal of side-chain protecting groups was performed in 95% trifluoroacetic acid (TFA, Sigma Aldrich), 0.5% Triisopropylsilane (TIPS, TCI Chemicals) for 30 min. Arrays were incubated twice in methanol for 30 s and rinsed four times with reagent-grade water (Ricca Chemical Co.). Arrays were washed for 1 min in TBST (1× TBS, 0.05% Tween-20), washed twice for 1 min in TBS, and exposed to a final wash for 30 s in reagent-grade water.

Serum samples were diluted 1:100 in binding buffer (0.01 M Tris-Cl, pH 7.4, 1% alkali-soluble casein, 0.05% Tween-20) and bound to arrays overnight at 4 °C. After sample binding, the arrays were washed three times in wash buffer (1× TBS, 0.05% Tween-20), 10 min per wash. Primary sample binding was detected via Alexa Fluor® 647-conjugated goat anti-human IgG secondary antibody (Jackson ImmunoResearch #109-605-098). The secondary antibody was diluted 1:10,000 (final concentration 0.1 ng/μl) in the secondary binding buffer (1× TBS, 1% alkali-soluble casein, 0.05% Tween-20). Arrays were incubated with secondary antibody for 3 h at room temperature, then washed three times in wash buffer (10 min per wash), washed for 30 s in reagent-grade water, and then dried by spinning in a microcentrifuge equipped with an array holder. Fluorescent signal of the secondary antibody was detected by scanning at 635 nm at 2 μm resolution and 15% gain, using an MS200 microarray scanner (Roche NimbleGen). Scanned array images were analysed with proprietary Roche software to extract fluorescence intensity values for each peptide.

**Data analysis: normalisation, quality control and removal of outliers (smoothing)**
**CHAGASTOPE-v1—quality control.** All experiments were performed in duplicate (same biological sample, duplicate assays on independent array slides). We performed quality control of each pair of replicates for each sample using Bland–Altman (MA) plots and reciprocal signal plots. All replicate array assays showed excellent overall reproducibility (see Supplementary Fig. S1). As another step of quality control, we analysed the replicas of the positive controls we placed in the array (these were sections of known antigenic proteins that had their peptides repeated four times in the microarray design). For this, we used their normalised signals (see below), and we observed excellent overall reproducibility between the replicas, both intra- and inter-array (see Supplementary File S10).

**CHAGASTOPE-v1—quantile normalisation.** To compare data across experiments, we normalised array data using quantile normalisation. Because this method requires similar statistical properties of the underlying distributions, we performed two sets of quantile normalisations, one for the assays using Chagas-positive samples, and one for the assays using negative samples (including those from leishmaniasis-positive serum samples, which produced signal distributions similar to the Chagas-negative samples). We treated replicas as independent samples, resulting in a normalisation across 12 array sets for the Chagas-positive samples and a normalisation across 16 array sets for the rest. Normalisation was performed in R using *normalize.quantiles* from the package *preprocessCore*[66].

**CHAGASTOPE-v1—smoothing and replicas.** To remove outliers, we used two methods combined: a rolling median smoothing procedure and an average via replicas. First, we assigned the normalised signal of each peptide to the corresponding protein sequences. This was done once per serum sample per replica. Next, we used the *rollmedian* function in the R package *zoo* to calculate the rolling median along each protein sequence. We used a window size of 3, meaning that the smoothed signal for each peptide was the median of itself and its two neighbouring peptides in the same protein/serum/replica (for peptides at the edges of the protein sequences, we added a 0 as the signal of the non-existing neighbouring peptide). Next, we combined data from the two replicas to calculate their average and standard deviation, resulting in the final dataset that was analysed and described herein. In this final dataset, each peptide had 14 associated signal values (6 from Chagas-positive samples, 6 from Chagas-negative samples, 1 from a leishmaniasis-positive sample and 1 from a leishmaniasis-negative sample). These signals can be found in Supplementary File S11.

**CHAGASTOPE-v2—quality control, quantile normalisation and smoothing.** In the CHAGASTOPE-v2 arrays, we followed similar steps as in the first design. Quality control: we analysed each pair of replicates for each sample using Bland–Altman (MA) plots and reciprocal signal plots (see Supplementary Fig. S3). Quantile normalisation: in these 12-plex assays, one microarray slide contained 12 sectors, which were assayed separately; hence, for all data-analysis purposes, 1 array sector was treated as one 1 array dataset. Quantile normalisation was thus performed for 142 assays (2 replicas for each of 71 individual serum samples). Smoothing and replicas: the smoothing and combining of replicas was done exactly as in CHAGASTOPE-v1 and the signals can also be found in Supplementary File S11.

**Data analysis: definition of antigenic peaks and regions**
**CHAGASTOPE-v1—antigenicity threshold.** To define this threshold, we analysed the normalised signals (before smoothing) for peptides in the 2 replicas of each of the 6 Chagas-positive and the 6 Chagas-negative pooled samples (totalling 24 samples) and calculated their mode and standard deviation. We then looked at the protein profiles (the smoothed signals) and analysed the dispersion of the 'noise', meaning how high the signals were for the healthy pools. Because this was a discovery screening, we wanted to be very conservative in this choice, making sure to select regions that were truly antigenic. All this, coupled with the amount of space available in our second design, led to us using an antigenicity threshold of 10,784.80 arbitrary fluorescence units (mode plus 4 standard deviations).

**CHAGASTOPE-v1—shared antigenic peptides.** To calculate the proportion of non-redundant antigenic peptides shared between two sera (in this case, pooled serum samples), we obtained the list of non-redundant antigenic peptides for each of them. We calculated the proportion as the number of non-redundant peptides those two lists had in common divided by the total number of non-redundant

peptides amongst those two lists (meaning, the intersection of the two sets divided by the union of the two sets).

**CHAGASTOPE-v1—peaks.** For the discovery screening, we defined as 'peak' a group of two or more consecutive peptides with signals greater than the antigenicity threshold. Because we were interested in the discovery of *T. cruzi* antigens and epitopes, we also required each peak to have a maximum signal in a Chagas-positive sample that was at least five times higher than the corresponding maximum signal in the negative samples for those peptides.

**CHAGASTOPE-v1—regions.** Antigenic regions result from the merging of neighbouring peaks. For each peak, we noted the position in its protein of the first and last peptides. We then expanded this range by moving the start of the peak 16 amino acids to the left and the end of the peak 16 to the right to ensure capturing the whole peak. Then, if two or more of these new 'wide peaks' overlapped with each other, they were all merged into one. This resulted in 9547 antigenic regions across both proteomes.

**CHAGASTOPE-v1—clusters of regions.** In our analysed proteomes, there were identical proteins or different proteins sharing significant sequence similarity over a domain or defined sequence region. This was either because they belonged to the same protein family or because the protein was present in both CL-Brener (Esmeraldo and Non-Esmeraldo haplotypes) and Sylvio X10 proteomes. This similarity resulted in several antigenic regions with very similar or identical sequences, which can distort the conclusions drawn from the data. To reduce redundancy, we clustered antigenic regions by sequence similarity using *blastp* (BLAST 2.2.31+)[67]. The all vs all comparison across all 9547 regions was run with the following *blastp* command options and parameters:

```
-outfmt '6 qseqid sseqid pident length mismatch gapopen
evalue bitscore qseq sseq sstart send' -word_size 2 -comp_
based_stats 0 -max_target_seqs 50000 -matrix BLOSUM80
```

We kept only matches with a percentage of identical amino acids (pident) of at least 80% and a match length of at least 75% of the length of the shortest region in the match. Using these matches, we computed a distance matrix where the distance was 1 – (pident/100) and then applied a single-linkage hierarchical clustering method. The resulting tree was cut at a cutoff of 0.2 (1 – pidentThreshold), resulting in 3868 Distinct Antigenic Regions.

**CHAGASTOPE-v1—clusters with regions similar to known antigens.** To analyse the novelty of identified antigens, we compared the sequences of all 9547 antigenic regions found in CHAGASTOPE-v1 against a list of 2243 known *T. cruzi* epitopes obtained from the Immune Epitope Database (IEDB)[68]. This list was obtained by selecting in IEDB: Epitope: 'Linear peptide'; Assay: only 'B Cell' and 'Positive'; Organism: '*Trypanosoma cruzi* (ID:5693)'; MHC Restriction: 'Any'; Host: 'Human'; Disease: 'Any'.

We used BLAST to compare this list against our antigenic regions. *blastp* was run using described parameters (see above), but we only kept matches with a percentage of identical amino acids (pident) of at least 80% and a match length of at least 50% of the length of the shortest sequence in the match. These matches can be seen in Supplementary File S12. We tagged a cluster as 'having regions similar to known antigens' if at least 10% of its regions matched with a known antigen (which in most cases meant at least 1 of them, see Supplementary Fig. S2).

**CHAGASTOPE-v2—antigenicity threshold.** To determine the antigenicity threshold in this experiment, we set to recreate the results

obtained in the discovery screening using virtual sample pools. The signal of a virtual pool for a given peptide was the highest signal for that peptide amongst the individual sera that were part of that pool (and were now being analysed individually). We then compared the antigenicity of these virtual pools against the antigenicity from our original pools in all clusters of antigenic regions using ROC curves. The goal was to predict the antigenicity in the original pools using the information from the corresponding virtual pools. An original pool was antigenic if it surpassed the 10,784.80 threshold, but for the virtual pools, we analysed possible thresholds of the formula mode plus $X$ standard deviations, where $X$ ranged from 1 to 4 in steps of 0.1 (using the mode and standard deviation from the second design). In the end, the best threshold was 5814.81 (mode plus 2.4 standard deviations) with an AUC of 0.83.

**CHAGASTOPE-v2—shared antigenic peptides.** Same as for CHAGASTOPE-v1.

**CHAGASTOPE-v2—comparison between strains.** To define if a peptide belonged to CL-Brener or Sylvio X10 (or to both), we looked at each of the peptides from the antigenic regions present in CHAGASTOPE-v2 and mapped these to their cognate proteins. Peptides that were mapped to proteins in both CL-Brener and Sylvio X10 were excluded from this analysis. Next, for each individual sample, we counted the number of non-redundant reactive peptides that were exclusive for each strain. When calculating the signal for the non-redundant peptides, we kept the highest signal between the peptides with the same sequence (there was more than one signal per peptide due to the smoothing). Because the CL-Brener proteome is larger (almost double the size of the Sylvio X10 proteome) we standardised the number of reactive non-redundant peptides using two sets of $Z$-scores, one for each strain.

While we performed this analysis using all 71 individual samples from CHAGASTOPE-v2, the interpretation of the reactivity observed for the individual sera containing '_E' in their sample name (those not used on the proteome-wide v1 arrays) has to be done with care, because these 38 serum samples were not used to select peptides for inclusion in the CHAGASTOPE-v2 arrays.

**CHAGASTOPE-v2—single-residue mutational scannings.** From the peaks found in the CHAGASTOPE-v1 primary screen, we selected those with antigenicity in at least two pools and a maximum signal of at least 21,500 fluorescence units (these parameters were a consequence of the space assigned in our second design for this experiment). This resulted in 1445 antigenic peaks from 977 proteins. For each of these peaks, we selected its best peptide and added its 4 neighbouring amino acids (2 to each side). This resulted in a set of sequences of 20 amino acids, of which 789 were distinct sequences.

In this study, we analysed all these sequences, although some different peaks are actually very similar with only residue variants. Therefore, and to clarify the presentation, peaks were clustered by similarity into 232 significantly different sequences. To accomplish this, we first calculated similarity across sequences using BLOSUM80 and the function parSeqSim from R package 'protr'[69]. Then, we grouped similar sequences in clusters, trying all possible numbers of clusters and selecting the optimal number according to a silhouette analysis scanning the performance. This resulted in 144 clusters which were later manually curated, where some clusters were splitted while others were merged, resulting in the final 232 significant sequences. For each cluster, we only present the result for the most antigenic peak sequence.

Mutational scanning of candidate epitopes was performed on a set of 5 overlapping peptides (16mers), which in concert cover a sequence of 20 amino acid residues. This is explained visually in Fig. 7 and in Supplementary Fig. S5. The best 16mer (max signal) in a selected

antigenic peak was labelled as the central peptide (position 0), then the 4 additional neighbouring peptides were labelled as their position relative to the central peptide, meaning positions −2, −1, +1, and +2. This resulted in 2914 unique peptides. The antibody-binding signal of each of these 5 peptides corresponds to the non-mutated (wild-type) versions of an epitope. To analyse the contribution of each residue in this sequence to the binding of specific antibodies, we generated all possible alanine variants (i.e., the replacement of each original amino acid in the sequence for alanine) from those 5 original peptides. In case the original amino acid was an alanine, glycine was used as a replacement. Since the peptides were 16mers (16 residues of length), we derived 80 mutated sequences for each peak, which added to the 5 original peptides resulting in a total of 85 peptides for each peak. A total of 45,519 unique peptides were generated for this mutational analysis.

Supplementary Fig. S5 shows a schematic view of this procedure for one antigen. Each residue from each peak sequence was analysed individually. The difference between the mean signal of the peptides that contained the non-mutated (or original) residue and those that contained the residue replaced by alanine or glycine, as appropriate, was calculated for each serum or several. This difference in each residue defines a profile change. See Supplementary File S13 for the full results of the example of Supplementary Fig. S8. An arbitrary cutoff point was established at 1300 to highlight residues with the greatest absolute change. Following this, sera were grouped according to the signal change profile of core residues (mean change > 1300). Finally, a sequence logo was constructed for visualisation purposes where the height of the amino acid is directly proportional to the signal change. This was created using a modified version of the R package 'ggseqlogo'[69] that can be found at: https://github.com/trypanosomatics/ggseqlogo.

**Synthesis and production of identified antigens in FLISA serologic assays.** For validation and assessment of diagnostic performance, identified antigens with no matches against the IEDB were produced as either short synthetic peptides or recombinant proteins. All produced antigens were named using the CAR-Ag (Chagas-Antibody-Repertoire-Antigens) prefix: CAR-Ag1, CAR-Ag2, CAR-Ag5, CAR-Ag9, CAR-Ag12, and CAR-Ag24 (Supplementary Data S10). Peptides were synthesised at Schafer-N ApS (Copenhagen, Denmark) using standard Fmoc-chemistry and were HPLC-purified (>85% purity). All peptides carry an N-terminal 12mer sequence (PS-tag) for increased binding to polystyrene plates[70]. Genes encoding recombinant antigens CAR-Ag1 and CAR-Ag24, containing N-terminal 6xHis tags, were synthesised and cloned at Genscript (NJ, USA), into pET21a+ expression vectors, and transformed into BL21(DE3) E. coli cells. Expression from a single positive colony was done on ZYM-5052 autoinduction medium[71], and purified to electrophoretic homogeneity using a combination of Ni-NTA affinity chromatography and ion-exchange chromatography. Antigens were assayed individually in a fluorescent-linked immuno-sorbent assay (FLISA) in half-area black polystyrene plates (Greiner BioOne, Cat. 675077). Plates were coated O.N. with individual CAR-Ag antigens (peptides: 250 ng/well, recombinant proteins 20 ng/well) in Phosphate Buffered Saline and blocked with 5% skim-milk in Tris Buffered Saline. Each serum sample was assayed in duplicate at 1:50 (Mexico cohort) or 1:100 (ARSD serodiscordant samples) dilutions, and detected using Alexa Fluor® 488-conjugated goat anti-human IgG secondary antibody (Jackson ImmunoResearch #109-545-098). Resultant fluorescence (λexc/ems = 485/535 nm) was measured using a FilterMax F5 Multimode microplate reader (Molecular Devices, CA, USA). Additional wells, coated with PBS or with the 12mer PS-tag peptide, were used as blanks in each plate for recombinant or peptide CAR-Ags, respectively. Signal from these wells was subtracted from the signal produced by antigen-coated wells and was expressed as Relative Fluorescence Units (RFU). The assay threshold (optimal cutoff) was

estimated based on Youden's J statistic[72]. As shown in Supplementary Fig. S15, we observed a good signal correlation when translating from microarray to 96-well FLISA assays. We also performed the Chagatest ELISA recombinant 4.0 (Wiener Labs, Argentina) on all samples following the instructions from the manufacturer.

### Reporting summary

Further information on research design is available in the Nature Portfolio Reporting Summary linked to this article.

## Data availability

The Peptide Array data generated in this study (raw and processed) have been deposited in the ArrayExpress Functional Genomics Data Collection under accession codes A-MTAB-692 and A-MTAB-693 (Array Designs); and under accession codes E-MTAB-11651 and E-MTAB-11655 (Assay Data). The processed peptide data mapped to *T. cruzi* gene identifiers, antigenicity plots for all proteins and Supplementary Files S1 to S13 are available at Figshare Collection: https://doi.org/10.6084/m9.figshare.19991021. Peptide array data can also be explored interactively at the https://chagastope.org website. The data that support the findings of this study are available without restrictions from the sources listed above. Source Data for all data presented in graphs within the manuscript are provided in the Source Data file. Source Data are provided with this paper.

## Code availability

Custom software used for data analysis is available at this GitHub Repository: https://github.com/trypanosomatics/The-Chagas-Disease-Antigen-and-Epitope-Atlas, https://doi.org/10.5281/zenodo.7696856. This software was created in R (4.0.3) and requires the following packages (dependencies): data.table (1.13.0), zoo (1.8-9), preprocessCore (1.52.1), dplyr (1.0.2), reshape2 (1.4.4) and pheatmap (1.0.12).

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

## Acknowledgements

The authors would like to acknowledge Santiago J. Carmona (Swiss Bioinformatics Institute, University of Lausanne) for critical reading of the manuscript; Javier M. Di Noia (Montreal Clinical Research Institute) for helpful suggestions; the Centro Regional de Chagas, Hospital Independencia, Santiago del Estero, Argentina, Luz María Peverengo, Luz María Rodeles, Diego Mendicino and Renato Uncos for technical support; and Laboratorios Wiener for the generous donation of Chagatest ELISA tests. Research reported in this publication was supported by the

National Institute of Allergy and Infectious Diseases (NIAID) of the National Institutes of Health under award number R01AI123070 (to F.A.), by an award from the Brockman Medical Research Foundation (to M.S.N.), by the National Agency for the Promotion of Science and Technology (ANPCyT, Argentina) under award numbers PICT-2013-1193, PICT-2017-0175 (to F.A.), and PICT-2019-03201 (to J.D.M.), and by the National Research Council of Argentina (CONICET), under award number PIP-0906 (to J.D.M.). The content is solely the responsibility of the authors and does not necessarily represent the official views of the National Institutes of Health.

## Author contributions

Conceptualisation: A.D.R., L.B., F.A. Data curation: A.D.R., L.B., F.A. Formal analysis: A.D.R., L.B. Funding acquisition: J.A., F.A. Investigation: A.D.R., L.B., E.S.-S., F.A. Methodology: A.D.R., L.B., E.S.-S., F.A. Project administration: F.A. Resources: E.S.-S., J.M.R., F.T., N.K., M.S.N., J.C.V., J.A., J.D.M., I.S.M., F.A. Software: A.D.R., L.B. Supervision: F.A. Validation: A.D.R., L.B., E.S.-S., F.A. Visualisation: A.D.R., L.B., F.A. Writing—original draft: A.D.R., L.B., F.A. Writing—review and editing: A.D.R., L.B., J.M.R., M.S.N., F.A.

## Competing interests

The authors declare no competing interests.
