## [Peer Review File · Nature Communications]

Reviewers' Comments:

Reviewer #1:

Remarks to the Author:

1) The research is technologically advanced and in that context is publishable. There are a few aspects that could be more clearly addressed: the serological biomarkers are not entirely novel; the survey may not be considered comprehensive, in that only two strains of *Trypanosoma cruzi* are used and initially pools of sera; the deliverables from the mutational scanning could be clarified; the descriptions of research impact and future work are limited.

2)

450 delete the word 'deliberately'

546: the authors here use a numeric citation system, but this numeric system does not seem to be used in the manuscript references, nor supplements.

3a)

The authors should state and make quite clear that several of the epitopes they identify in Table 2 are highly similar to already defined antigens, namely:

The epitope from TCSYLVIO_009094 and TcE identified by Houghton et al (1999)

The epitope from TcCLB.511633.79 and Pep36 (Ag36) identified by Ibanez et al 1988 (and subsequently re-identified as JL9 (Levin et al 1989) and MAP (Kerner et al 1989))

The epitope from TCSYLVIO_002712 and Pep2 (Ag2) identified by Ibanez et al 1988 (and subsequently re-identified as TCR39 (Hoft et al 1989) and B13 (Gruber & Zingales 1993))

The epitope from TcCLB.508325.230 and Pep13 (Ag13) identified by Ibanez et al 1988 (and subsequently re-identified as TcD (Burns et al 1992))

3b)

Furthermore, the authors should make quite clear that some of these antigens are already well known and are in fact used in the commercial tests that the authors describe as applied to these samples (lines 581-590), namely:

Chagatest ELISA recombinant v4.0 (Argentine samples): Pep2 (Ag2); Pep13 (Ag13); Pep36 (Ag36)
Chagas Stat-Pak (USA samples): B13.

Thus, the selection of the samples may have an element of bias.

Reviewer #2:

Remarks to the Author:

In this manuscript, the authors draw a serum antibody reactivity atlas for Chagas disease patients. They do so by first selecting important reactive peptides (potential epitopes) from pooled serum samples in order to then measure binding reactivities individually. This is important work as Chagas disease is a Neglected disease. The work performed is extensive and well described. I would, however, like a more quantitative and understandable analysis of the data...the figures produced right now seem very low-dimensional given the data is so high-dimensional. Please see my comments below.

Major

- What's the variation of peptide binding across patients?? Fig 5 shows counts...but what's the variation of signal intensities? And what's the pairwise overlap of peptide reactivities across individual samples? I would like to see a heatmap of all individuals...

- What's the signal intensity of a monoclonal antibody with known peptide reactivity compared to serum reactivity signal intensities? What I am asking is, how high is the signal intensity of a positive control compared to serum antibody reactivities?

- Overall, figures are very difficult to interpret, I would like to see figures more easily accessible. For example, fig 4 is almost incomprehensible to me.

- The peptides that made it to the second array, how similar are they? And are there all more similar compared to the entire proteome? In other words, are the peptide targeted by the immune system a connected subnetwork of the entire proteome?

Minor

"Although the molecular mechanisms that produce diverse antibody repertoires are precisely understood (Di Noia and Neuberger, 2007; Neuberger, 2008)" → please rephrase since incorrect.

Reviewer #3:

Remarks to the Author:

Agüero and co-workers reported a study, purely based on peptide microarray data, where they analyzed antibody specificities in Chagas Disease patients across the Americas. First, microarrays of 2,441,908 unique peptides (overlapping 16-mer peptides with an offset of 4 amino acids) spanning two *T. cruzi* proteomes (hybrid CL Brener & Sylvio X10 strains) were screened with six different (positive) patient serum pools (4–6 samples/pool) and six (control) negative serum pools, as well as a *Leishmania* positive/negative control for cross-reactivity tests. The authors detected 21,062 reactive unique peptides (conservative threshold of ~10k AFU) in 7,707 proteins, with about 3,868 independent clusters of antigenic regions (if similarity-sorted by BLAST). They conclude that the immune responses in Chagas disease patients are highly diverse and only few epitopes are conserved (publicly shared) across many patients, which has also been hypothesized and confirmed by others in various single-celled parasites. 166 antigenic regions were shared by at least 4 of the analyzed pooled samples and only 43 were reactive in all pooled samples (5 of these were cross-reactive in *leishmania* samples).

A second, more focused screening was performed with 241,772 unique peptides selected from "all" binders + some surrounding non-binders of the first screen (now with 15 amino acid overlap), as well as including some alanine scan substitutions of 232 antigenic regions (peaks). 71 individual patient samples (33 from the previous pools + 38 new samples) were screened on 144 array replicas (each sample as duplicate). Confirming the previous results using pooled sera, the 71 individual patient samples showed that most clusters of antigenic regions were detected by a low number of individuals (85 % of antigenic clusters show seroprevalence of < 20 %). Thus, the patients show a large repertoire of private antigens. Here, the threshold was already reduced to ~5k AFU, which results in a higher seroprevalence of antigenic regions. Finally, the authors investigate the individual specificities of the patients towards the antigenic regions and discuss the impact of alanine scan on the antibody binding modes. This part is quite interesting, but more speculative and should be discussed a bit more carefully (see minor comments).

In summary, this is an outstanding peptide microarray study, giving a comprehensive antigenic map of the *T. cruzi* (linear!) peptidome, diving deep into the array data. Good statistical analysis, normalization, and stratification were performed. Yet, I am highly ambivalent about this manuscript for this journal: While I enjoy such screening analyses very much, the impact at the current state seems rather limited. Some major concerns arise, which have to be addressed in a manuscript at this level for Nature Communications:

1. Application in diagnostics and disease research: While this is a great dataset, what is the main application of this? There are several ELISA tests available for Chagas disease. If these supposedly novel markers have diagnostic or disease research value, these need to be validated/investigated with other approaches. Validation of this data with other techniques, such as pre-synthesized and spotted peptides on microarrays or peptide ELISA has to be performed. While data and presentation are credible, unfortunately, it is not convincing enough to only present microarray data. The top ~40 binders should be analyzed/validated with other assays with additional serum samples. Either show that some of these correlate with certain disease parameters (development of immunity, severity, etc.) or show their improved diagnostic value. What is their performance versus other commercial ELISA assays? If this shows (even only minor) improvement → publish this in Nature Communications. Otherwise, this work is much more suitable for a specialized journal (Communications Biology).

As another example of how difficult translation of microarray data can be: Please compare your study to Mishra et al., mBio 2018 (<https://journals.asm.org/doi/full/10.1128/mBio.00095-18>), where they found a promising peptide biomarker for Zika, using peptide microarrays. Yet, it was extremely difficult to translate this into an assay (e.g. ELISA). Concluding, identifying potential biomarkers on microarrays is only the first step! There is much more work to be done.

2. Context & novelty: How many new antigenic clusters were identified? Or is this work mainly confirming known antigens? Please give a comprehensive list with literature comparisons, especially of the publicly targeted antigenic regions (with high seroprevalence).

"While some of these regions were already known T. cruzi antigenic proteins, there were novel ones identified, which are of interest for improving diagnosis and treatment of Chagas disease." -> Be specific! How many of these regions were already known? How many were newly identified? This is crucial for the novelty of this work (e.g. also compared to the author's own work: Carmona, 2015). And: How many of the known antigenic regions could the authors not identify, supposedly due to linear peptide approach? If this is not sufficiently addressed, it gives an impression that most of the data have already been published previously.

Concluding: A biological study, which only relies on a single technique without validation or novel findings/applications is in my opinion insufficient to be published in Nature Communications. What is the use of this data beyond this study? The main finding is a multitude of highly individual/private epitopes and only limited epitopes/biomarkers with high seroprevalence. This is well known for several diseases and does not justify high-impact publication. Furthermore, typical conformational epitopes are neglected/not found due to the technology restrictions (as stated in the author's limitations). Quite often, diagnostic and/or neutralizing antibodies exclusively bind to 3d/conformational epitopes.

Microarray experiments are typically just the start of a study or to support a finding. Although major work has been done (which is impressive!), to publish this in Nature Communications, a large database from peptide microarrays is not convincing enough (lack of novelty!). Is there any specific gain or unexpected outcome? I have not seen (or missed?) this in the manuscript.

Minor concerns:

1. Why did the authors use 16-mer peptides, while many other studies using the same technology (photolabile protecting groups) had large problems with synthesis quality, restricting to 12-mer peptides (e.g. Mishra 2018 and many others)? Can the authors confirm in a small analysis that binding epitopes especially when analyzing overlapping peptides sliding over a short epitope that the synthesis quality is sufficient (e.g. short 4, 5, or 6-mer epitope sliding from C- to N-terminus in overlapping peptides is still bound, regardless of position within peptide in comparable intensity)?
2. P. 16 Line 386ff "...at least 7 individuals showed increased antibody-binding...": Be careful with such interpretation of the data. An alanine scan does not necessarily prove an increase in binding! Only a strong or complete loss of signal may be correlated with importance of the amino acid residue for binding. An increase might simply represent an increase in synthesis yield. Such data would need to be verified in other assays.
3. A conservative threshold of ~10k AFU for pooled samples will only give very strong or common binders in one sample. This was selected to disregard 99% of peptides for the second patient-individual screening. Potential loss of many signals, bias, patient grouping/stratification can cause large bias/problems!
4. Please use only one style: serums or sera (preferably the second one) for the plural of serum
5. The number of samples per pool should be included in Fig. 1, since it varies between 4 – 6 patients per pool.

Reviewer #4:

Remarks to the Author:

In this manuscript, called "A Trypanosoma cruzi Antigen and Epitope Atlas: deep characterization of antibody specificities in Chagas Disease patients across the Americas", the authors describe the proteome-wide identification of linear epitopes and antigenic regions in the proteomes of 2 T. cruzi strains. Chagas disease, the disease caused by this parasite, is a neglected tropical disease with an annual incidence of 30,000 new cases and leading to 12,000 deaths per year.

The authors have used high-density peptide arrays in their work and although it is not the first time that this technology is being used to map the linear epitope repertoire of a parasite, it is the first time, to my knowledge, that it has been used for *T. cruzi*. Additionally, the authors have performed a thorough bio-IT analysis of the data obtained from these arrays, making it a rich and interesting source of data for both diagnostic development and possibly also vaccine development.

The study has been well conducted, with sufficiently large sample sets and a second array to confirm the initial data was used, further increasing the confidence in the data set. One shortcoming of this study is however that none of the identified linear epitopes have been confirmed using an independent technology such as ELISA or Luminex. While I understand this might be out of scope for this manuscript, I would encourage the authors to include this in the Discussion section.

I would be in favor of publishing this work, but do have a number of comments I'd like to see addressed:

1. The authors speak in the abstract and in other parts of the work about 'novel high seroprevalence antigens' and 'public epitopes'. Both terms relate in fact to the more commonly used terminology 'immunodominant epitopes and antigens'. It might be better to use this term throughout the manuscript.

2. Two strains of *T. cruzi* are used to maximize the display of relevant peptide variants with only two genomes. Why haven't the authors used a more data driven approach, similar to what was used by Francesca Falconi-Agapito (Frontier in Immunology, 2021) in the context of dengue virus to cover all known lineages? Additionally, it would be good to see the difference between both proteomes in a graphical way or at least give some more information about the sequence similarity at amino acid level and how that relates to the broader set of known proteome sequences.

3. Authors speak about a comprehensive atlas of antigens and epitopes (e.g. line 100-103) while in fact only linear epitopes have been investigated. This should be made more clear in the abstract and introduction as it is an important limitation of the methodology used.

4. In the discovery screening, also 2 pools of Leishmaniasis-negative and positive samples were included. However, in figure 2 these 2 pools are not presented. Might be good for transparency to also include these data here.

5. The threshold being used is indeed very conservative. It is much higher than what has been published by others (e.g. Lagatie et al., PLOS NTD 2017). Why was such a conservative threshold used? Additionally, in the M&M section it is described that his threshold was calculated based on the Chagas-positive samples. I would assume this was done on the Chagas-negative samples, to calculate the threshold based on noise?

6. Line 156 talks about cross-reactive peaks. I would not say these are cross-reactive peaks. They might simply not be related to Chagas at all, but might be caused by exposure to something else (in both positive and negative groups). I'd rather say these are non-specific peaks.

7. The authors have chosen to only look to sequence similarity using BLAST at the antigenic region level. However, it is known that linear epitopes often have only 4-5 essential residues, which will not be picked up by BLAST when using larger regions as input. Indeed, the authors find after the alanine scanning analysis that several peptides that were originally identified in fact contain the same epitope. Would it not have been better to already look to certain sequence motifs in the list of 16,737 antigenic peaks after Array 1 and in this way immediately identify a list of peaks that all relate to the same epitopes? In Table 2, I see for instance lines 8,9, 10 and 14 having all the same motif L/FRQIDx. The same is also true for Table 3.

8. The second array was used to analyze 71 serum samples. Unfortunately, the authors did not include any healthy control samples in this experiment. Additionally, a different threshold was used compared to the 1st array. This is a bit strange as the threshold should be the same (eventually

after normalization to allow for array-to-array variation).

9. In figure 5, only the 33 samples that were also used for discovery are displayed and not the 38 new samples. It is expected that the samples also used for discovery experiments give good results also in confirmation array. The new set of samples is essential as it demonstrates that not all peptides or regions will survive. I would therefore encourage to have figure 5 replaced by Suppl Figure S5.

10. Only 232 peptides were included in the Alanine scanning. They were selected based on the Discovery Array as they were included immediately in the second array. How did you select them? The way it is described now, it gives the impression that they were selected based on the confirmation array, which is not the case.

11. Line 365 please change overlapped to overlapping

12. lines 412-415 discuss about TCSYLVI0_002530 protein. The authors state that it is not an antigen because it in fact contains the same sequence as the antigenic repeat in Ag2/CA-2. This is not entirely correct. This discussion is about the difference between an antigen and an immunogen. Most likely in this case Ag2 is the immunogen that induces the immune response while the other protein is an antigen (meaning it is recognized by antibodies). Also, the authors say that it is a mimotope. A mimotope however, is a linear epitope that mimics the structure of a (conformational) epitope. This is not the case here. The only thing that can be said, is that the epitope sequence is present not only in the immunogen, but also in other proteins, which in nature might not be antigenic at all because the particular linear epitope sequence is not accessible for the immune system or the protein not expressed in a way the immune system can detect it...

13. Lines 476-480. The authors need to acknowledge that many other parasites/bacteria/viruses may also produce cross-reactive antibodies. this has not been studied as such now.

14. Lines 499-501 It would be good if the authors could elaborate a bit more on why it is of interest to have the dataset on T. cruzi peptides from healthy subjects. What is the value of these data?

15. For the first array, the authors have used different normalization for positive and negative samples. This is weird. One would expect to use the same normalization over all samples. I realize the distributions are different for both groups, but if one wants to use normalization, then it needs to be used over all samples. Otherwise, it does not really make sense to normalize.

RESPONSES TO REVIEWERS

Nature Communications manuscript

Original: NCOMMS-22-32606-T

Revised: NCOMMS-22-32606-A

We thank the reviewers for their time and helpful comments, which served to improve the manuscript greatly. We have responded in detail to each raised concern or comment (our responses in blue below), and have revised the manuscript accordingly.

REVIEWER COMMENTS

Reviewer #1 (Remarks to the Author):

1) The research is technologically advanced and in that context is publishable. There are a few aspects that could be more clearly addressed: the serological biomarkers are not entirely novel; the survey may not be considered comprehensive, in that only two strains of *Trypanosoma cruzi* are used and initially pools of sera; the deliverables from the mutational scanning could be clarified; the descriptions of research impact and future work are limited.

We thank the reviewer for this observation. To clarify this we have now improved the analysis and changed how we present results. Briefly, many of the serological biomarkers are novel; however, all previously described serological markers do also appear as positive hits in the screening and in the Atlas (with the exception of nonlinear epitopes, as noted in the text). We have perhaps stressed too much a number of known antigens in the examples and figures in the original submission, as these serve as a validation of the technique and provide support for the discovery of novel antigens.

In the revised Ms we have now tried to highlight in more detail which epitopes and antigens are novel, and which correspond to previously described ones. To classify antigens into novel vs known, we relied on sequence similarity to determine if antigenic regions were similar to validated/known antigens. Hence we performed BLASTP to compare the sequences of 9,547 antigenic regions against 2,243 epitopes found in the Immune Epitope Database (IEDB.org) and tagged our regions and clusters as previously known or not (see revised Methods for details). We used this information to modify Figure 6 (novel, includes previous Fig 6) and Supplementary Figure S2 (novel, includes previous Fig S4), and also used it to reselect which regions are shown in Tables 2 and 3, as well as adding this information to Supplementary Tables S5 (previous S6) and S7 and to Supplementary Files S2 and S5 (previous Sup Tables S12 and S13). We also added a new large table as Supplementary File S12 (novel), which has all the “good” matches between our regions and the IEDB epitopes.

*Regarding the coverage of the survey, we agree that it is the most comprehensive done to this day, but may not be comprehensive in a more strict sense. Yes we have only surveyed two strains of *T. cruzi* initially, but strain CL-Brener is a hybrid strain, derived from an ancestral genetic exchange (see Gaunt MW et al 2003). Hence, although strictly we did only survey two strains, the fact that CL-Brener has ~ 2x the number of genes of non-hybrid strains, and that these genes represent the two haplotypes/alleles of the ancestral strains that entered into the hybridization-based genetic exchange extend the coverage of our survey to ~ 3 extant haplotypes.*

Regarding the deliverables from the mutational scanning, we are not sure exactly what it is that the reviewer is asking. The main deliverable is information, that can be encoded in different ways, on the relevance of epitope residues for antibody binding (see Supp Table S8). This information can guide downstream development and improvement of diagnostic and other immunoassay reagents. For large-scale bioinformatic analysis (e.g. searching genomes), epitope information can be encoded as regular expressions (patterns), or as position-specific scoring matrices (PSSMs).

Regarding the description of research impact and future work, we have now revised this (based also on remarks by Reviewer #3). The revised manuscript now contains additional work showing impact of the discovered antigens (see new section “Novel antigens for improved diagnosis of Chagas Disease”, line 454).

2)

450 delete the word ‘deliberately’

546: the authors here use a numeric citation system, but this numeric system does not seem to be used in the manuscript references, nor supplements.

Thanks for these suggestions. We have corrected and fixed these.

3a)

The authors should state and make quite clear that several of the epitopes they identify in Table 2 are highly similar to already defined antigens, namely:

The epitope from TCSYLVIO_009094 and TcE identified by Houghton et al (1999)

The epitope from TcCLB.511633.79 and Pep36 (Ag36) identified by Ibanez et al 1988 (and subsequently re-identified as JL9 (Levin et al 1989) and MAP (Kerner et al 1989))

The epitope from TCSYLVIO_002712 and Pep2 (Ag2) identified by Ibanez et al 1988 (and subsequently re-identified as TCR39 (Hoft et al 1989) and B13 (Gruber & Zingales 1993))

The epitope from TcCLB.508325.230 and Pep13 (Ag13) identified by Ibanez et al 1988 (and subsequently re-identified as TcD (Burns et al 1992))

We agree and we understand the point the reviewer is making. However, based also on the comment #1 above by this same reviewer we have now decided to focus on novel antigens when presenting data in the main article. Hence we have now revised Tables 2 and 3 to only show novel antigens. We have also added information in other supplementary tables marking which regions are highly similar to already defined antigens in the Immune Epitope Database (new Supplementary Files S2, S5, and S12). The revised Figure 6 now shows in a clear way the classification of antigens into novel vs matching the IEDB. We also provide the IEDB identifiers that allow users to trace back each epitope to the original studies for all these antigens.

The new Supplementary File S12 lists epitope sequences and the corresponding match to IEDB epitopes and their descriptions. We also provide links to access the corresponding IEDB epitope pages which shows assays, publications, and other curated information for each antigen.

3b)

Furthermore, the authors should make quite clear that some of these antigens are already well known and are in fact used in the commercial tests that the authors describe as applied to these samples (lines 581-590), namely:

Chagatest ELISA recombinant v4.0 (Argentine samples): Pep2 (Ag2); Pep13 (Ag13); Pep36 (Ag36)

Chagas Stat-Pak (USA samples): B13.

Thus, the selection of the samples may have an element of bias.

Agreed. As explained above, we have now marked clearly in the revised manuscript (e.g. in Figure 6, Tables 2 and 3 and in Supplementary Tables and Files) which antigens match previously described antigens, and which are novel.

Regarding the selection of samples, yes, there is an unavoidable bias in their selection as these were selected firstly based on a positive serological diagnosis of infection using existing kits. Hence we expected to find reactivity against peptides derived from these antigens (and we did!), which served as positive support (see new Supplementary File S10). However, different clinical sites used different serology assays for diagnosis of infection (see Methods), hence there is also a level of diversity (reduced selection bias) in the selection of samples.

Reviewer #2 (Remarks to the Author):

In this manuscript, the authors draw a serum antibody reactivity atlas for Chagas disease patients. They do so by first selecting important reactive peptides (potential epitopes) from pooled serum samples in order to then measure binding reactivities individually. This is important work as Chagas disease is a Neglected disease. The work performed is extensive and well described. I would, however, like a more quantitative and understandable analysis of the data...the figures produced right now seem very low-dimensional given the data is so high-dimensional. Please see my comments below.

We thank the reviewer for providing a detailed critique of the work.

Yes, the figures were produced with the decision to lower their dimensionality so as to make them easy to consume by what we believe is the target audience of this work. For all purposes we have tried to produce figures that can guide scientists to the locations of epitopes on antigens, and to produce visual representations of seroprevalence, but we understand there is another audience and that figures depicting data in other ways may help illustrate the work better. Hence we have now produced other visualizations of the data (see new Figure 6, and Supplementary Figures S2a and S4, mentioned below).

Major

- What's the variation of peptide binding across patients?? Fig 5 shows counts...but what's the variation of signal intensities? And what's the pairwise overlap of peptide reactivities across individual samples? I would like to see a heatmap of all individuals...

To answer this question, we analyzed how antigenic peptides were shared between each pair of sera. These data was used to create two new heatmaps: Supplementary Figure S2a, showing the comparison between pooled sera from CHAGASTOPE-v1 arrays, and the new Figure 6c and Supplementary Figure S4, showing the comparison between individual sera from CHAGASTOPE-v2. We observed that the proportion of non-redundant antigenic peptides shared between two sera was below 30% for all comparisons. We also observed a natural grouping of chagas-positive, leishmania-positive and healthy serums in Supplementary Figure S2a. Interestingly we also observed a grouping of sera into Northern America + Northern South America vs Southern South America (see revised Figure 6 and Supplementary Figure S4). We thank the reviewer for the question, which helped improve the manuscript. We have now included this analysis into the main text and discussed these findings.

- What's the signal intensity of a monoclonal antibody with known peptide reactivity compared to serum reactivity signal intensities? What I am asking is, how high is the signal intensity of a positive control compared to serum antibody reactivities?

Agreed. Unfortunately we have not assayed any monoclonal antibody with these arrays. However, the signal intensity of a positive control compared to a serum from a chagas-negative serum can be observed globally in Supplementary Figures S1a (previous S2a, raw data) and S1b (previous S2b, normalized data). Here we present the distribution of signal intensities for all 2.84 million peptides in the arrays for pools of positive samples and matched negative samples.

Also, different positive controls produce antibody-signals in different signal ranges, e.g. positive peptides from the Ag2/CA-2 antigen (TcCLB.508831.140, Ibañez et al 1988); or the TSSA antigen (TcCLB.507511.81, Di Noia JM et al 2002) produce very high signals (when positive), and hence very large signal ratios when comparing positive vs negative samples (~ 50,000 relative units for CHAGASTOPE-v1 arrays, and ~ 40,000 in CHAGASTOPE-v2 arrays). Whereas peptides from other positive (known) antigens produce overall lower (though still very high) signals (e.g. TcCLB.507447.19, Ag36/MAP Ibañez et al 1988), and still others produce constantly positive signals (over the cutoff) with not very high signal (hence with lower positive vs negative signal ratios). These can be analyzed and observed at the companion chagastope.org website.

- Overall, figures are very difficult to interpret, I would like to see figures more easily accessible. For example, fig 4 is almost incomprehensible to me.

We apologize for this. We have now modified Figure 4 to make it more clear that it is showing three distinct examples of stacked antibody-binding profile sub-figures for the antigenic regions assayed in CHAGASTOPE-v2. We also modified the text in the main body and in the legend of the figure with the same objective in mind (lines 266 - 291).

- The peptides that made it to the second array, how similar are they? And are there all more similar compared to the entire proteome? in other words, are the peptide targeted by the immune system a connected subnetwork of the entire proteome?

The antigenic peptides are by definition a subset of the entire proteome, and also because the epitopes recognized by antibodies are present in different overlapped peptides in the arrays it is expected then that several peptides will be detected by antibodies present in a reactive (positive) serum sample (see for example Supp Figure S14 in the revised Ms). Also, because of their sequence overlap they will be more similar between each other than random peptides from a complete proteome. As explained in Methods, in CHAGASTOPE-v1 arrays two consecutive 16mer peptides in a protein sequence will be overlapped in 12 residues (hence they will be 75% identical). In CHAGASTOPE-v2 arrays, two consecutive 16mer peptides are overlapped in 15 residues (hence they are 94% identical). If an epitope is contained in ~ 7-8 consecutive peptides (e.g. see response to Reviewer #3 below) then we expect based on sequence similarity and contiguity in the protein sequence that these peptides will form a kind of "subnetwork". However this is an obvious subnetwork.

*Besides the overlapping in the tiling array design, another underlying reason for clustering of antigenic regions is the presence of gene families in the *T. cruzi* genome. If one member of a*

gene family is antigenic, and the reactive epitope is 100% conserved across other family members (irrespective of whether it may or may not be expressed or exposed to the immune system as part of these other family members), then the measured antibody binding in the arrays will be mapped back from peptides onto all the corresponding proteins. This may be one explanation for the observed presence of similar sequences clustered as ‘antigenic’. In the absence of perfect conservation of antigenic epitopes, the alternative explanation would be that different members of the gene family are all antigenic per se (i.e. expressed and exposed to the immune system). In this context, they may be clustered together depending on their level of sequence similarity, and these may also form a kind of “subnetwork”.

Besides these cases, after clustering most clusters are distinct enough to disprove the idea that antigenic peptides exist in some sort of single connected subnetwork. In summary, we do not find that the 3,868 clustered antigenic regions can be further connected without further relaxing the sequence similarity clustering parameters to produce non-biological significance levels.

Minor

.
“Although the molecular mechanisms that produce diverse antibody repertoires are precisely understood (Di Noia and Neuberger, 2007; Neuberger, 2008)” —> please rephrase since incorrect.

We apologize for this, we have now rephrased this sentence to make it more accurate. Now it reads “Although the cellular and molecular mechanisms that produce diverse antibody repertoires and underpin antibody affinity maturation in response to infection are largely understood (Bannard and Cyster, 2017; Di Noia and Neuberger, 2007; Feng et al., 2020; Zhang et al., 2022), a comprehensive description of their specificities in different infected individuals has been hindered by the lack of powerful tools.” (lines 61 - 65). The bibliographic references that provide support to this sentence were also updated to reflect current knowledge on these mechanisms.

Reviewer #3 (Remarks to the Author):

Agüero and co-workers reported a study, purely based on peptide microarray data, where they analyzed antibody specificities in Chagas Disease patients across the Americas. First, microarrays of 2,441,908 unique peptides (overlapping 16-mer peptides with an offset of 4 amino acids) spanning two *T. cruzi* proteomes (hybrid CL Brener & Sylvio X10 strains) were screened with six different (positive) patient serum pools (4–6 samples/pool) and six (control) negative serum pools, as well as a *Leishmania* positive/negative control for cross-reactivity tests. The authors detected 21,062 reactive unique peptides (conservative threshold of ~10k AFU) in 7,707 proteins, with about 3,868 independent clusters of antigenic regions (if similarity-sorted by BLAST). They conclude that the immune responses in Chagas disease patients are highly diverse and only few epitopes are conserved (publicly shared) across many patients, which has also been hypothesized and confirmed by others in various single-celled parasites. 166 antigenic regions were shared by at least 4 of the analyzed pooled samples and only 43 were reactive in all pooled samples (5 of these were cross-reactive in *leishmania* samples).

A second, more focused screening was performed with 241,772 unique peptides selected from “all” binders + some surrounding non-binders of the first screen (now with 15 amino acid overlap), as well as including some alanine scan substitutions of 232 antigenic regions (peaks). 71 individual patient samples (33 from the previous pools + 38 new samples) were screened on 144 array replicas (each sample as duplicate). Confirming the previous results using pooled sera, the 71 individual patient samples showed that most clusters of antigenic regions were detected by a low number of individuals (85 % of antigenic clusters show seroprevalence of < 20 %). Thus, the patients show a large repertoire of private antigens. Here, the threshold was already reduced to ~5k AFU, which results in a higher seroprevalence of antigenic regions. Finally, the authors investigate the individual specificities of the patients towards the antigenic regions and discuss the impact of alanine scan on the antibody binding modes. This part is quite interesting, but more speculative and should be discussed a bit more carefully (see minor comments).

In summary, this is an outstanding peptide microarray study, giving a comprehensive antigenic map of the *T. cruzi* (linear!) peptidome, diving deep into the array data. Good statistical analysis, normalization, and stratification were performed. Yet, I am highly ambivalent about this manuscript for this journal: While I enjoy such screening analyses very much, the impact at the current state seems rather limited. Some major concerns arise, which have to be addressed in a manuscript at this level for Nature Communications:

We thank the reviewer for the detailed review and for the comments. We provide responses below to each point.

1. Application in diagnostics and disease research: While this is a great dataset, what is the main application of this? There are several ELISA tests available for Chagas disease. If these supposedly novel markers have diagnostic or disease research value, these need to be validated/investigated with other approaches. Validation of this data with other techniques, such as pre-synthesized and spotted peptides on microarrays or peptide ELISA has to be performed.

While data and presentation are credible, unfortunately, it is not convincing enough to only present microarray data. The top ~40 binders should be analyzed/validated with other assays with additional serum samples. Either show that some of these correlate with certain disease parameters (development of immunity, severity, etc.) or show their improved diagnostic value. What is their performance versus other commercial ELISA assays? If this shows (even only minor) improvement -> publish this in Nature Communications.

Otherwise, this work is much more suitable for a specialized journal (Communications Biology). As another example of how difficult translation of microarray data can be: Please compare your study to Mishra et al., mBio 2018 (<https://journals.asm.org/doi/full/10.1128/mBio.00095-18>), where they found a promising peptide biomarker for Zika, using peptide microarrays. Yet, it was extremely difficult to translate this into an assay (e.g. ELISA). Concluding, identifying potential biomarkers on microarrays is only the first step! There is much more work to be done.

Yes, we agree with the reviewer, and indeed we have been working already on top binders. This experimental work was already ongoing during submission and reviewing of this Ms. We prioritized and produced ~ 35 antigens for further study using diverse criteria (some of which fall outside the scope of this work). We do understand the added value and impact of the additional translational work, hence we're now happy to include some of these results. For chronic Chagas disease (diagnosis of infection) we have synthesized ~ 14 top binders as either short synthetic peptides, or produced recombinant proteins (in those cases where there are several antigenic epitopes in the same antigen, spread over the length of the protein). We have successfully validated most binders in FLISA format (fluorescent-linked immunosorbent assays), and we demonstrate improved diagnostic performance in ELISA format for Chagas Disease for 6 of them in the revised manuscript for two collections of challenging samples (see new Figure 8).

In the new Figure 8a we show the reactivity of 3 of these antigens against a panel of 17 serodiscordant samples. Serodiscordance in Chagas disease is a recognized problem (see Guzmán-Gomez D et al 2015; Moure Z et al 2016; Castro Eiro MD et al 2017). As explained in the Introduction, the WHO guidelines for diagnosis of Chagas Disease recommend two serological tests, and a third test if results are discordant. In the revised manuscript we included the analysis of 17 samples (see Supplementary Table S10, rows 14 - 30) that gave discordant results in ELISA vs IHA and were classified into Chagas positive (likely infected) or negative (likely non-infected) by a third test (IIF) performed in a different reference center by trained personnel. The 3 novel antigens CAR-Ag1, CAR-Ag12, and CAR-Ag24 showed clear reactivity against serodiscordant samples with positive IIF outcome (likely infected), and very low reactivity against serodiscordant samples with negative IIF outcome (likely non-infected). In concert 6 out of the 12 IIF positive samples (50%) were detected by these antigens and are thus very promising antigens for the development of improved diagnostic reagents.

And in Figure 8b we show the reactivity of 4 novel antigens against a panel of 168 samples from Mexico (see Supplementary Table S10, rows 31 - 198). As we now explain in the revised introduction, there is emerging evidence of variations in test sensitivity in different human populations, with particularly high rates of discordance between serological tests in Mexico or US T. cruzi infections with Mexican origins (Umezawa ES et al 1999; Sosa-Estani S et al 2008; Verani JR et al 2009; Guzmán-Gomez D et al 2015). As shown in the figure the new antigens show very good performance in FLISA assays against these samples. Particularly, one of these novel antigens alone (CAR-Ag1) outperforms a commercial diagnostic reagent (Wiener Recombinant v4.0, which is one of the few FDA-approved Chagas tests, and is also one of the most sensitive tests for North-American samples (see Kelly EA et al 2021) for this particular collection of samples (see new Table 4 in the revised manuscript).

2. Context & novelty: How many new antigenic clusters were identified? Or is this work mainly confirming known antigens? Please give a comprehensive list with literature comparisons, especially of the publicly targeted antigenic regions (with high seroprevalence).

“While some of these regions were already known T. cruzi antigenic proteins, there were novel ones identified, which are of interest for improving diagnosis and treatment of Chagas disease.”

-> Be specific! How many of these regions were already known? How many were newly identified? This is crucial for the novelty of this work (e.g. also compared to the author’s own work: Carmona, 2015). And: How many of the known antigenic regions could the authors not identify, supposedly due to linear peptide approach? If this is not sufficiently addressed, it gives an impression that most of the data have already been published previously.

Agreed. This is also a comment by Rev #1. We have addressed this and clarified which were previously known T. cruzi antigenic proteins. This is explained in the responses to Rev#1, but we have now revised the Ms to put more emphasis on novel antigens and have also identified and labeled peptides and antigenic regions similar to previously identified antigens. In summary, we modified Figure 6 (novel, includes previous Fig 6) and Supplementary Figure S2 (novel, includes previous Fig S4), we used this new information to reselect which regions are shown in Tables 2 and 3, and we added this information to Supplementary Tables S5 (previous S6) and S7, as well as to Supplementary Files S2 and S5 (previous Sup Tables S12 and S13). Finally, we also added a new large table as Supplementary File S12 (novel), which has all the “good” matches between our regions and the IEDB epitopes (see response to Reviewer #1 and new Methods for details).

Concluding: A biological study, which only relies on a single technique without validation or novel findings/applications is in my opinion insufficient to be published in Nature Communications.

What is the use of this data beyond this study? The main finding is a multitude of highly individual/private epitopes and only limited epitopes/biomarkers with high seroprevalence. This is well known for several diseases and does not justify high-impact publication. Furthermore, typical conformational epitopes are neglected/not found due to the technology restrictions (as

stated in the author's limitations). Quite often, diagnostic and/or neutralizing antibodies exclusively bind to 3d/conformational epitopes.

Microarray experiments are typically just the start of a study or to support a finding. Although major work has been done (which is impressive!), to publish this in Nature Communications, a large database from peptide microarrays is not convincing enough (lack of novelty!). Is there any specific gain or unexpected outcome? I have not seen (or missed?) this in the manuscript.

We agree and expect that in the revised version of the Ms we are convincing the reviewer that there is additional validation on the identified epitopes, and that there are specific gains when using these antigens to improve diagnosis of Chagas Diseases in different contexts: serodiscordant samples and specific human populations where current tests show limitations.

Minor concerns:

1. Why did the authors use 16-mer peptides, while many other studies using the same technology (photolabile protecting groups) had large problems with synthesis quality, restricting to 12-mer peptides (e.g. Mishra 2018 and many others)? Can the authors confirm in a small analysis that binding epitopes especially when analyzing overlapping peptides sliding over a short epitope that the synthesis quality is sufficient (e.g. short 4, 5, or 6-mer epitope sliding from C- to N-terminus in overlapping peptides is still bound, regardless of position within peptide in comparable intensity)?

Good question, and yes we can confirm this. We have obviously taken great care in analyzing and validating the peptide array platform for quite some time already and have confidence in the quality of synthesis. As a more specific response to your question, we have now added in Supplementary Materials a number of examples that show this (see new Supplementary Figure S14). Specifically, for short epitopes where we know the extent of the epitope from the mutagenesis analysis (Alanine scan), we can validate that sustained high antibody-binding signal is observed irrespective of the location of the epitope within the 16mer peptides. In these cases we have not seen any suggestion of problems during synthesis or quality issues, as epitopes located at the end (respective to peptide synthesis direction) still show very strong antibody-binding signals.

2. P. 16 Line 386ff "...at least 7 individuals showed increased antibody-binding...": Be careful with such interpretation of the data. An alanine scan does not necessarily prove an increase in binding! Only a strong or complete loss of signal may be correlated with importance of the amino acid residue for binding. An increase might simply represent an increase in synthesis yield. Such data would need to be verified in other assays.

Indeed. And we have verified this in other assays. In the heatmaps we produced for alanine-scan analysis of epitopes, the increase in antibody binding is i) observed across different arrays; ii) only for some serum samples; and iii) it is the result of measuring binding in 5 different peptides where this residue was mutated (see Supplementary Figure S5 in the revised Ms (previous Figure S6)) and contrasting these measurements against those in 5 other non-mutated peptides. In concert this means that when we observe both increased and decreased antibody binding for that residue in different samples it is difficult to explain as an increase in synthesis yield. Please see the example below (in particular the glutamic residue E39). Although strictly we cannot discard an issue with synthesis, we believe this is unlikely, as the pattern of decreased vs increased binding of mutated peptides is non-random.

3. A conservative threshold of ~10k AFU for pooled samples will only give very strong or common binders in one sample. This was selected to disregard 99% of peptides for the second patient-individual screening. Potential loss of many signals, bias, patient grouping/stratification can cause large bias/problems!

This is only partially true. Yes, we have used a very conservative threshold for pooled samples, and yes this will give strong binders in each sample. This was done based on our previous experience with other peptide arrays (Carmona SJ et al 2012; Carmona SJ et al 2015), and in our experience in translating array reactivity to other assay platforms (e.g. ELISA, see Mucci J et al 2017).

*However selection of candidates for the design of the second screening was done at the level of peaks, not individual peptides. Hence a **single peptide** giving a high signal above the ~10k threshold would be considered an outlier and be removed. However a peak is defined by many peptides over the threshold. Regarding the loss of many signals, this is also partly true, as signals below the cutoff in one pool of samples may be above the cutoff in another pool. Hence the fact that selection was performed at the peak-level (per sample) prevented incurring into selection bias. The fact that this led to removal of ~ 99% of the peptides was a consequence rather than part of the criteria for design of the second array.*

The results from the patient-individual screening (see old Figure 6, now Figure 6a in the revised Ms) also support a lack of selection bias. The fact that we observe a large fraction of low seroprevalence antigens (85% of the antigenic regions were only reactive in <20% of individuals) in the patient-individual screening, similar to what we observed in the pooled-samples screening, shows that removal of peptides did not affect the distribution of seroprevalences (as observed in Figure 3). We further provide here some visual examples to support this:

Antigen TcCLB.508335.9 (CL-Brener DGF-1) showed a single positive peak in microarrays assayed with a pool of US samples (see below). No reactivity was observed with other samples (AR, BO, BR, CO, MX). This did not prevent this peak from being selected for the patient-individual screening. Subsequently, this peak was reactive (at different levels) in the individual-patient screening in patient samples from Colombia (see below), Mexico (see below), Bolivia, Brazil and US samples (data available for re-analysis and also at the companion chagastope.org website).

We do understand that a hard limit meant that other peaks from this antigen (below the cutoff, see peaks below at positions ~500 and ~950) were not included in the secondary screening (the same could happen for other antigens). However, we do not find that this selection introduced any significant bias as explained.

Data from pooled samples (screenshots from companion chagastope.org website):

Data from individual samples (screenshots from companion chagastope.org website):

Regarding our use of a very conservative threshold, we worked under the assumption that positives coming out of this screening would be translated into other immunoassays, e.g. multiwell immunosorbent assays. We have previous experience in translation of candidate peptides from different peptide array platforms to ELISA assays (Carmona SJ et al 2012; Carmona SJ et al 2015; Mucci J et al 2017). We agree with the reviewer that translation of reactive peptides in microarrays to e.g. ELISA assays may be hard or difficult, however in our experience we have been largely successful in moving in this direction using either the best peptide in a peak or other nearby peptides. This does not mean that we have not failed. We usually synthesize and assay only one peptide per antigen, and often these short peptides will be reactive against a few sera but will fail to reproduce the expected level of seroprevalence. So success when translating to our FLISA assays is not straightforward (yes the peptide was reactive, no the peptide did not achieve the expected performance). But by fine mapping antibody-binding peaks at high-resolution (1-residue offset); and having some assessment of quantitative signal plus seroprevalence (population diversity), we have successfully selected and produced antigens either as recombinant proteins or as synthetic peptides as explained above. Below we are providing a couple of examples of our experience when moving from arrays to FLISA assays.

Here we can see correlation of signals between microarrays and fluorescent-linked immunosorbent assays (FLISA). The plots show one recombinant protein (CAR-Ag1) and one short synthetic peptide (CAR-Ag9). FLISA reactivity is the average of duplicate assays. Microarray reactivity is the cumulative signal of all individual peptides that span the antigen used in FLISA assays (e.g. all peptides that span the region of the antigen that make up the recombinant CAR-Ag1 protein or all 16mer peptides in the array that span the sequence of the synthetic CAR-Ag9 peptide).

Of course these reagents can be further improved. As the data in the paper and supp materials show that there is diversity in how antibodies from different patients bind their cognate epitopes (single-residue mutational scanning), hence, we expect that the best 16mer for one subject may be a suboptimal peptide for another patient. The data in this Ms can certainly serve as a foundation to guide us and others in future work to improve these reagents.

4. Please use only one style: serums or sera (preferably the second one) for the plural of serum

Thanks! Corrected throughout the Ms.

5. The number of samples per pool should be included in Fig. 1, since it varies between 4 – 6 patients per pool.

Agreed, we added this information to Figure 1.

Reviewer #4 (Remarks to the Author):

In this manuscript, called "A Trypanosoma cruzi Antigen and Epitope Atlas: deep characterization of antibody specificities in Chagas Disease patients across the Americas", the authors describe the proteome-wide identification of linear epitopes and antigenic regions in the proteomes of 2 T. cruzi strains. Chagas disease, the disease caused by this parasite, is a neglected tropical disease with an annual incidence of 30,000 new cases and leading to 12,000 deaths per year.

The authors have used high-density peptide arrays in their work and although it is not the first time that this technology is being used to map the linear epitope repertoire of a parasite, it is the first time, to my knowledge, that it has been used for T. cruzi. Additionally, the authors have performed a thorough bio-IT analysis of the data obtained from these arrays, making it a rich and interesting source of data for both diagnostic development and possibly also vaccine development.

The study has been well conducted, with sufficiently large sample sets and a second array to confirm the initial data was used, further increasing the confidence in the data set. One shortcoming of this study is however that none of the identified linear epitopes have been confirmed using an independent technology such as ELISA or Luminex. While I understand this might be out of scope for this manuscript, I would encourage the authors to include this in the Discussion section.

We thank the reviewer for the detailed review and for the comments. We provide responses below to each point. Regarding the identified shortcoming (which was also pointed out by Reviewer #3), we have now expanded the Manuscript to validate antigens in a multiwell immunosorbent assay and show their diagnostic potential on challenging samples.

I would be in favor of publishing this work, but do have a number of comments I'd like to see addressed:

1. The authors speak in the abstract and in other parts of the work about 'novel high seroprevalence antigens' and 'public epitopes'. Both terms relate in fact to the more commonly used terminology 'immunodominant epitopes and antigens'. It might be better to use this term throughout the manuscript.

We appreciate the clarification on the terminology. Our microarrays show that certain peptides can be detected by antibodies found in a larger proportion of individuals. We refer to these peptides and epitopes as having 'high seroprevalence'. We understand the concept of immunodominance but because we are using seroprevalence as an operational measure (see for example lines 457-471 in the revised manuscript), we find that the concept of seroprevalence (number or fraction of individuals reacting with an epitope) is easier to convey than immunodominance. We do agree with the reviewer that using the terminology 'public epitopes' may be confusing with other uses, so we removed that terminology from the Ms.

2. Two strains of *T. cruzi* are used to maximize the display of relevant peptide variants with only two genomes. Why haven't the authors used a more data driven approach, similar to what was used by Francesca Falconi-Agapito (Frontier in Immunology, 2021) in the context of dengue virus to cover all known lineages? Additionally, it would be good to see the difference between both proteomes in a graphical way or at least give some more information about the sequence similarity at amino acid level and how that relates to the broader set of known proteome sequences.

*We appreciate the comment by the reviewer. Regarding the data driven approach, we must highlight the fact that while the Dengue virus has a small genome (~10-11 Kb, coding for a single polyprotein of 3391 residues or 7 shorter mature proteins), the *T. cruzi* genomes are larger (~ 55-60 Mb each; ~ 10 thousand proteins per genome). Regarding the request of visualizing the differences between proteomes, we did indeed perform a thorough preliminary research on the diversity of encoded peptides using available genomes before embarking on the design of our v1 discovery arrays. This was not described in this Ms, but for completeness we are now providing here the data that we obtained back then (there are now many more complete genomes available that were not part of this analysis). The figure below illustrates the diversity of short peptides obtained from one genome (strain CL-Brener which was the first *T. cruzi* genome sequenced, and is the reference genome), and how this diversity increases when adding additional genomes. We also indicate the evolutionary lineage or DTU (Discrete Typing Unit) for each strain (TcVI, TcI, TcII, etc). This shows that there is a significant degree of diversity covered by the CL-Brener + Sylvio X10 genomes. The next available genome back then was from the Dm28c strain which belongs to the same evolutionary lineage as Sylvio X10 (TcI). Taken into consideration both this fact and the array capacity (~ 3 million peptides) we decided to use these two genomes to explore antibody specificities in Chagas disease subjects. Also as explained in the Ms, the CL-Brener genome (as other genomes from the TcVI DTU) is a hybrid genome containing ~2X the number of genes of other strains. This genome contains the haplotypes and alleles from ancestral TcII and TcIII strains that led to the appearance of hybrid TcVI strains (see Zingales B et al 2012).*

3. Authors speak about a comprehensive atlas of antigens and epitopes (e.g. line 100-103) while in fact only linear epitopes have been investigated. This should be made more clear in the abstract and introduction as it is an important limitation of the methodology used.

*We have now clarified that this is all about linear epitopes. We have updated the abstract, that now reads: "We describe the first proteome-wide search for antigens, characterised their **linear** epitopes, and show their reactivity on 71 individuals from diverse human populations". The last paragraph of the introduction now reads "As a result, we produced a comprehensive atlas of antigens and their **linear** epitopes, providing a unique resource for understanding adaptive immune responses against this parasite". The first sentence in the Discussion already mentioned this in the original submission (it has now been changed slightly, but the reference to linear epitopes didn't change): "We have produced a detailed map of the antibody specificities against **linear** epitopes in patients with Chagas disease, described in depth here for the first time". Also, the Discussion still contains a complete paragraph (lines 560 - 567) explaining that the microarray platform is biased towards detection of binding to short peptides, hence only detecting linear epitopes.*

4. In the discovery screening, also 2 pools of Leishmaniasis-negative and positive samples were included. However, in figure 2 these 2 pools are not presented. Might be good for transparency to also include these data here.

Agreed. This was now added to Figures 1 and 2.

5. The threshold being used is indeed very conservative. It is much higher than what has been published by others (e.g. Lagatie et al., PLOS NTD 2017). Why was such a conservative threshold used? Additionally, in the M&M section it is described that his threshold was

calculated based on the Chagas-positive samples. I would assume this was done on the Chagas-negative samples, to calculate the threshold based on noise?

Yes. This concern is similar to one raised by Reviewer #3, we have addressed this issue above.

When reviewing this question we realized we had not mentioned the use of negative (healthy) subjects as part of the calculation of the antigenicity threshold (this has been fixed in the revised Methods). Regarding this calculation, we have used both negative + positive data, yes. The reason for this is that in both datasets ~99% of the peptides in the CHAGASTOPE-v1 arrays (pooled serum samples) were non-reactive (even for the Chagas-positive serums). This meant that the statistical mode in both cases was very similar (70 vs 68). In CHAGASTOPE-v2 arrays this rationale was not valid anymore (there was now a larger proportion of antigenic – positive – peptides in the microarray), so we changed how we calculated our threshold for these arrays as explained in Methods, and also below.

6. Line 156 talks about cross-reactive peaks. I would not say these are cross-reactive peaks. They might simply not be related to Chagas at all, but might be caused by exposure to something else (in both positive and negative groups). I'd rather say these are non-specific peaks.

Agreed, and corrected. Thanks!

7. The authors have chosen to only look to sequence similarity using BLAST at the antigenic region level. However, it is known that linear epitopes often have only 4-5 essential residues, which will not be picked up by BLAST when using larger regions as input. Indeed, the authors find after the alanine scanning analysis that several peptides that were originally identified in fact contain the same epitope. Would it not have been better to already look to certain sequence motifs in the list of 16,737 antigenic peaks after Array 1 and in this way immediately identify a list of peaks that all relate to the same epitopes? In Table 2, I see for instance lines 8,9, 10 and 14 having all the same motif L/FRQIDx. The same is also true for Table 3.

Yes, this is correct. We initially used BLAST to find and group similar antigenic regions. Regarding the mutational analysis, yes we find some similar mutational profiles in different proteins, and yes this provides support for the idea of using motifs. However, while this is easily done once a motif is defined (e.g. L/FRQIDx, then use this motif to find other proteins containing it), performing de novo motif identification from an heterogeneous list of sequences (reactivity against different non-related epitopes) is not straightforward. We did attempt a preliminary strategy using the MEME suite (for de novo motif identification using expectation-maximization, Bailey TL et al 2015) but the results were very noisy and difficult to put in biological context.

*To further clarify, we believe that if we would have assayed the arrays with a monoclonal antibody, or a highly enriched polyclonal serum (e.g. one obtained after immunization with a well defined antigen), then yes, the resulting list of reactive peptides could have been fed to a de novo motif detection and we would likely identify the set of residues involved. Further, if in addition to this, we would have designed the arrays in a different way, perhaps including more sequence variants either from natural sources (genomes from other *T. cruzi* isolates) or performing full mutational scans (not just alanine), then again yes, de novo motif identification would be an obvious strategy. However in our situation we have lists of antigenic regions reactive against polyclonal sera containing many different epitopes and motifs. We believe that the deconvolution of many embedded motifs in the data even for a single serum sample is a complex computational problem.*

8. The second array was used to analyze 71 serum samples. Unfortunately, the authors did not include any healthy control samples in this experiment. Additionally, a different threshold was used compared to the 1st array. This is a bit strange as the threshold should be the same (eventually after normalization to allow for array-to-array variation).

Yes, the second array design was not assayed with negative/control samples. Regarding the different threshold, this is because the two arrays used slightly different slides. The first arrays are single-sector slides (meaning that ~ all of the available surface is used to display peptides and for the immunoassay; a single sample is assayed per slide; and the size of the peptide fields (aka spots) is larger). However the arrays used for the second design are 12-plex arrays where the available surface of the slide is divided in 12 assayable sectors + divider space (not usable). In these arrays the size of the peptide fields is smaller. In concert all these differences meant that we could not assume that the signal range would be the same across experiments. That was the reason why we took the extra effort to analyze the signals of the same set of positive peptides against the same set of samples used in both v1 and v2 arrays to determine the antigenicity threshold used in our second array. This is described in Methods (lines 752-764 in the original submission).

9. In figure 5, only the 33 samples that were also used for discovery are displayed and not the 38 new samples. It is expected that the samples also used for discovery experiments give good results also in the confirmation array. The new set of samples is essential as it demonstrates that not all peptides or regions will survive. I would therefore encourage to have figure 5 replaced by Suppl Figure S5.

Agreed, we have now replaced Figure 5 with the previous Supplementary Figure S5. In this revised Figure we also sorted samples by difference in reactivity against unique peptides from one strain or the other. We believe this now helps readers visualize the differences better.

10. Only 232 peptides were included in the Alanine scanning. They were selected based on the Discovery Array as they were included immediately in the second array. How did you select them? The way it is described now, it gives the impression that they were selected based on the confirmation array, which is not the case.

*Yes, these were selected from the Discovery Arrays. This was clear in the Methods, but in the main text it was unfortunate that the ordering of the sections in the text had the mutational scanning analysis included at the end of the manuscript, after having discussed and presented results from the v2 arrays. This may be confusing and have now slightly rewritten the introductory text to this section, which now reads: "The CHAGASTOPE-v2 array design **already** included mutants designed in silico for 232 selected antigenic peaks found in the discovery screening."*

11. Line 365 please change overlapped to overlapping

Fixed. Thanks!

12. lines 412-415 discuss about TCSYLVIO_002530 protein. The authors state that it is not an antigen because it in fact contains the same sequence as the antigenic repeat in Ag2/CA-2. This is not entirely correct. This discussion is about the difference between an antigen and an immunogen. Most likely in this case Ag2 is the immunogen that induces the immune response while the other protein is an antigen (meaning it is recognized by antibodies). Also, the authors say that it is a mimotope. A mimotope however, is a linear epitope that mimics the structure of a (conformational) epitope. This is not the case here. The only thing that can be said, is that the epitope sequence is present not only in the immunogen, but also in other proteins, which in nature might not be antigenic at all because the particular linear epitope sequence is not accessible for the immune system or the protein not expressed in a way the immune system can detect it...

Agreed. This is the correct way of describing and we have taken the suggestion to rewrite this paragraph (lines 428 - 439 of the new Ms). Thanks!

13. Lines 476-480. The authors need to acknowledge that many other parasites/bacteria/viruses may also produce cross-reactive antibodies. this has not been studied as such now.

Agreed. We have added a sentence to the Discussion that reads "Cross-reactivities to other parasites, bacteria or viruses have not been studied and will have to be explored when validating these epitopes." (lines 573 - 575 of the new Ms).

14. Lines 499-501 It would be good if the authors could elaborate a bit more on why it is of interest to have the dataset on T. cruzi peptides from healthy subjects. What is the value of these data?

Agreed. The value of these data is that it provides information on precision-mapped linear epitopes from antigens that are non-specific for Chagas Disease. These are reactive against normal human sera, from different human populations. Irrespective of the nature of the immunogen (natural exposure to other pathogens or immunogens) they may serve as controls in future large scale immunomics experiments. As discussed in the manuscript (lines 596 - 614 of the new Ms), one application of these biomarkers for Chagas Disease, is in the study of dynamics of these antibody responses as a result of perturbation of the host-parasite interaction (e.g. by chemotherapeutic treatment and reduction of parasite loads in the host). In this context, one may want to observe and quantify the presence of antibodies against both specific as well as non-specific epitopes. We propose that information on these non-specific epitopes will be important in the design of these large scale experiments.

15. For the first array, the authors have used different normalization for positive and negative samples. This is weird. One would expect to use the same normalization over all samples. I realize the distributions are different for both groups, but if one wants to use normalization, then it needs to be used over all samples. Otherwise, it does not really make sense to normalize.

Yes this is correct. We have used the same normalization technique (quantile normalization) but have applied this separately for both classes (positive vs negative). The reasons for this is that the distributions are different for both groups as mentioned by the reviewer. Quantile normalization works under the assumption that the underlying distribution is the same across classes, and this does not hold in this case. Hence we disagree that it needs to be the same over all samples. Also see for example Zhao, Wong and Bin Goh (2020), <https://www.nature.com/articles/s41598-020-72664-6>.

Reviewers' Comments:

Reviewer #1:

Remarks to the Author:

Thank you for providing the responses from the reviewers and the authors.

The original submission had some significant problems, because it should have been made clear that several antigens are already well-known and are used in diagnostic tests, and this biased the analyses. The authors have now recognised this and as far as possible have made some adjustments.

The dependence of the study on only two strains still has constraints, even though one is hybrid.

The technological approach is novel and is interesting.

The initial deliverables/impact of the research were not strong, as also mentioned by reviewer 3; in response the authors have now added more data and have strengthened the discussion.

As stated, with the improvements, in my opinion the work is publishable.

Reviewer #2:

Remarks to the Author:

The authors have addressed all of my comments.

Reviewer #3:

Remarks to the Author:

In this revised manuscript, the authors have presented very convincing arguments and have done extensive work for validation. The authors now clearly show the novelty and impact of their work: (1) Comparing their identified epitopes with the literature (IEDB) and, (2) validation with another assay (FLISA).

Overall, very well-done work. Congratulations!

Presentation of the data is much clearer and validation with FLISA assay is convincing (new Fig. 8, Table 4), although at its infancies. It is somewhat unfortunate that not only short peptides, but also larger proteins need to be engineered, to get better FLISA results (and cover longer epitopes in one assay, Table 4). Yet, this was expected and the strategy was also successfully followed by others. It is also quite interesting that most of the high-seroprevalence antigens were already found previously (Fig. 6B). New Figure 6C is also quite informative.

Selection bias and synthesis yield problems were well addressed. Thank you! The yield issue may be discussed further in other settings.

Minor suggestions:

1. I would very much appreciate, if the authors would include the two graphs from their rebuttal, comparing "FLISA reactivity vs. cumulative signal on microarray" for CAR-Ag1 & 9 in the SI with a bit more explanation.
2. The SI should be corrected for "serums" to "sera", to comply with the main manuscript.

Reviewer #4:

Remarks to the Author:

In this revised version of the manuscript entitled "The Trypanosoma cruzi Antigen and Epitope Atlas: antibody specificities in Chagas disease patients across the Americas", the authors have diligently addressed the comments that were provided by the different reviewers.

I appreciate the additional work that has been included on peptide ELISA confirmation of the identified peptides. This is definitely an added value for the manuscript.

I would be in favor of publishing this version of the manuscript.

RESPONSES TO REVIEWERS

Nature Communications manuscript

Original: NCOMMS-22-32606-A

Revised: NCOMMS-22-32606-B

We thank all reviewers for their time and helpful comments, which served to improve the manuscript greatly. We are responding below to the final comments and suggestions (our responses in blue below), and have revised the manuscript accordingly.

REVIEWER COMMENTS

Reviewer #1 (Remarks to the Author):

Thank you for providing the responses from the reviewers and the authors.

The original submission had some significant problems, because it should have been made clear that several antigens are already well-known and are used in diagnostic tests, and this biased the analyses. The authors have now recognised this and as far as possible have made some adjustments.

The dependence of the study on only two strains still has constraints, even though one is hybrid.

The technological approach is novel and is interesting.

The initial deliverables/impact of the research were not strong, as also mentioned by reviewer 3; in response the authors have now added more data and have strengthened the discussion.

As stated, with the improvements, in my opinion the work is publishable.

We thank Reviewer #1 for the comments, and are glad that you find the work and deliverables provides more strength and impact.

Reviewer #2 (Remarks to the Author):

Reviewer #2 (Remarks to the Author):

The authors have addressed all of my comments.

We thank Reviewer #2 for the comments and are glad that we have addressed your concerns.

Reviewer #3 (Remarks to the Author):

In this revised manuscript, the authors have presented very convincing arguments and have done extensive work for validation. The authors now clearly show the novelty and impact of their work: (1) Comparing their identified epitopes with the literature (IEDB) and, (2) validation with another assay (FLISA).

Overall, very well-done work. Congratulations!

Thanks! We thank Reviewer #3 for the comments which helped improve the manuscript and are glad that we have addressed your concerns.

Presentation of the data is much clearer and validation with FLISA assay is convincing (new Fig. 8, Table 4), although at its infancies. It is somewhat unfortunate that not only short peptides, but also larger proteins need to be engineered, to get better FLISA results (and cover longer epitopes in one assay, Table 4). Yet, this was expected and the strategy was also successfully followed by others. It is also quite interesting that most of the high-seroprevalence antigens were already found previously (Fig. 6B). New Figure 6C is also quite informative.

Selection bias and synthesis yield problems were well addressed. Thank you! The yield issue may be discussed further in other settings.

Yes, sometimes short peptides work reasonably well, and in the case of the larger proteins we engineered, this was mostly done to save costs. Otherwise we could have synthesized many short peptides and used them together to coat wells for FLISA assays (as we have done in Mucci J et al 2017). In the cases of antigens with many epitopes, where there were also differences in reactivity of these epitopes across patients, we felt that using a recombinant protein containing the antigenic region (spanning several epitopes) would be a sensible strategy. And it worked for us in these cases.

Regarding the high prevalence antigens, yes many were already found previously but most were not. The majority of antigens are novel, and we are disclosing those in this paper.

Minor suggestions:

1. I would very much appreciate, if the authors would include the two graphs from their rebuttal, comparing “FLISA reactivity vs. cumulative signal on microarray” for CAR-Ag1 & 9 in the SI with a bit more explanation.
2. The SI should be corrected for “serums” to “sera”, to comply with the main manuscript.

We appreciate the suggestion. We have now included this figure as a new Supplementary Figure (S15), with an appropriate explanation and legend, and have fixed the use of serums. Thanks!

Reviewer #4 (Remarks to the Author):

In this revised version of the manuscript entitled "The Trypanosoma cruzi Antigen and Epitope Atlas: antibody specificities in Chagas disease patients across the Americas", the authors have diligently addressed the comments that were provided by the different reviewers.

I appreciate the additional work that has been included on peptide ELISA confirmation of the identified peptides. This is definitely an added value for the manuscript.

I would be in favor of publishing this version of the manuscript.

We thank Reviewer #4 for the comments and are glad that you find the manuscript has improved. Thanks!